Manuscript prepared for Atmos. Meas. Tech.
with version 3.0 of the LaTeX class copernicus.cls.
Date: 12 June 2019

# A geometry-dependent surface Lambertian-equivalent reflectivity product at 466 nm for UV/Vis retrievals: Part I. Evaluation over land surfaces using measurements from OMI

Wenhan Qin[1], Zachary Fasnacht[1], David Haffner[1], Alexander Vasilkov[1], Joanna Joiner[2], Nickolay Krotkov[2], Bradford Fisher[1], and Robert Spurr[3]

[1]Science Systems and Applications Inc., Lanham, MD, USA
[2]NASA Goddard Space Flight Center, Greenbelt, MD, USA
[3]Rt Solutions Inc., Cambridge, MA, USA

*Correspondence to:* Wenhan Qin (wenhan.qin@ssaihq.com)

**Abstract.** The anisotropy of the Earth's surface reflection has implications for satellite-based re­trieval algorithms that utilize climatological surface reflectivity databases that do not depend upon the observation geometry. This is the case for most of the current ultraviolet and visible (UV/Vis) cloud, aerosol, and trace-gas algorithms. The illumination-observation dependence of surface reflec­tion is described by the bidirectional reflectance distribution function (BRDF). To account for the BRDF effect, we use the concept of geometry-dependent surface Lambertian-equivalent reflectiv­ity (GLER), which is derived from the top-of-atmosphere (TOA) radiance computed with Rayleigh scattering and surface BRDF for the exact geometry of a satellite-based pixel. We present details on the implementation of land and water surface BRDF models, and evaluate our GLER product over land surfaces using observed sun-normalized radiances at 466 nm. The input surface BRDF parameters for computing TOA radiance are derived from MODerate-resolution Imaging Spectro­radiometer (MODIS) satellite observations. The observed TOA radiance for comparison is from the Ozone Monitoring Instrument (OMI). The comparison shows good agreement between observed and calculated OMI reflectivity in 2006 in typical geographical regions, with correlation coefficients greater than 0.8 for some regions. Seasonal variations of clear-sky OMI reflectivity (i.e., with min­imum clouds and aerosols) closely follow those computed using MODIS-derived GLER over land. GLER also captures the cross-track dependence of OMI-derived LER, though the latter is slightly higher than the former presumably owing to residual cloud and aerosol (non-absorbing) contamina­tion, particularly over dark surfaces (heavily vegetated regions such as mixed forest, croplands and grasslands). Calibration differences between OMI and MODIS may also be responsible for some of this bias. The standard OMI climatological surface reflectivity database predicts higher radiances

than GLER and OMI observations with different seasonal variation over most regions and does not have any angular-dependent variation. Overall, our evaluation demonstrates that the GLER product adequately accounts for surface BRDF effects while at the same time simplifies the surface BRDF implementation within the existing OMI retrieval infrastructure; use of our GLER product requires changes only to the input surface reflectivity database.

## 1 Introduction

It is well-known that reflection of the incident sunlight by the Earth's surface is generally anisotropic in the optical wavelength range (Rencz and Ryerson, 1999). Rough surfaces (vegetated landscapes, urban and built-up, bare soils, etc) usually exhibit marked backward scattering, whereas smooth surfaces (e.g., water, snow/ice) tend to have a strong forward scattering peak (specular reflection). Two well-known phenomena related to surface reflection anisotropy are the hot-spot effect over land and the sunglint over ocean. The hot-spot effect occurs when the viewing direction coincides with the illumination direction, such that all shadows are invisible. This results in a reflectance peak in backward scattering directions (e.g., Qin et al., 1996). Sunglint, however, is a peak in forward scattering caused by Fresnel reflection over a smooth surface such as calm water, when sunlight reflects off the surface at the same angle that the surface is viewed (e.g., Kay et al., 2009).

The dependence of surface reflection on illumination and observation directions is mathematically described by the bidirectional reflectance distribution function (BRDF), an intrinsic property of the surface (Nicodemus, 1965; Martonchik et al., 2000; Schaepman-Strub et al., 2006). Since BRDF is defined in terms of differential solid angles, in theory it cannot be measured (Nicodemus, 1977). Therefore, another quantity which can be retrieved from remote sensing data, the bidirectional reflectance factor (BRF), has been widely used ever since. BRF is defined as the ratio of the reflected radiance from the surface to that from a perfect Lambertian surface under the same geometry (illumination and observation) and ambient conditions. Since an ideal diffuse surface reflects the same radiance in all viewing directions, the BRDF for a Lambertian surface is $1/\pi$. Because of this, the BRF for any surface is equal to its BRDF times $\pi$. However, unlike the BRDF, BRF is a unitless quantity.

The effect of surface anisotropy on satellite-observed radiances in the visible is notable and neglect of it in retrievals can produce complex errors. The influence of surface anisotropy on the top of the atmosphere (TOA) radiance increases with wavelength for a Rayleigh atmosphere (no aerosols or clouds) because atmospheric transmittance increases with wavelength in the ultraviolet and visible (UV/Vis) spectral regions. The radiation incident on the surface consists of a direct component (non-scattered radiation) and a diffuse component scattered by the atmosphere (gases, aerosols, and clouds). The magnitude and spectral distribution of the diffuse irradiance depends on atmospheric conditions. Over a clear sky, the diffuse component originates from Rayleigh scattered sunlight that

follows a $\lambda^{-4}$ dependence, where $\lambda$ is wavelength. As a result, the surface anisotropy's impact on TOA radiance is strong at visible or longer wavelengths because the atmosphere is more transparent than in the UV where Rayleigh scattered light is more prominent and therefore smooths and reduces the surface BRDF effect at UV wavelengths. Obviously, the longer the wavelength, the stronger the effects, as shown in Lorente et al. (2018) when comparing surface anisotropy effects in the near-infrared (NIR) with that in the visible.

The surface reflectance anisotropy has implications for UV/Vis satellite retrievals of aerosols, clouds, and trace gases such as nitrogen dioxide ($NO_2$). Currently, most satellite-based UV/Vis algorithms (e.g., Krotkov et al., 2017) use surface reflectivity climatologies, typically gridded monthly Lambertian-equivalent reflectivities (LERs) that have been derived from satellite observations, for example, Herman and Celarier (1997) from the Total Ozone Mapping Spectrometer (TOMS) at 340 and 380 nm, Koelemeijer et al. (2001) from the Global Ozone Monitoring Experiment (GOME) in 11 wavelengths between 335-772 nm, Kleipool et al. (2008) from OMI in 23 wavelengths at 328-499 nm, and more recently Tilstra et al. (2017) from GOME-2 in 21 wavelengths between 335-772 nm as well as from the Scanning Imaging Absorption Spectrometer for Atmospheric Chartography (SCIAMACHY) in 29 wavelengths from 335-1670 nm. These climatologies are constructed by computing statistical values representative of multiple years of observations made with different sun and viewing geometries. In order to minimize cloud contamination, they may be based on a lower percentile (e.g., Herman and Celarier, 1997) and/or the mode of the LER histogram depending on surface type (e.g., Koelemeijer et al., 2001; Kleipool et al., 2008; Tilstra et al., 2017). As pointed out recently by Lorente et al. (2018), such climatologies tend to pick up the lowest values among the measurements of a scene, typically corresponding with forward (backward) scattering geometries over land (water).

An example of the impact of LER climatologies on cloud fraction retrievals is the presence of considerable cross-track biases. This has been shown for satellite retrievals in the $O_2$-A band (Wang et al., 2008) as well as for the 477 nm $O_2-O_2$ band (Veefkind et al., 2016). This happens because the LER climatologies tend to underestimate the actual LER in the backward scattering directions over land since the hot-spot phenomenon is not represented properly and the retrieval compensates for this by overestimating cloud fractions in order to match the observed TOA reflectance. Over ocean, such overestimation of cloud fraction would occur in the forward scattering direction due to neglect of sun glint.

To account for surface anisotropy in existing cloud and trace gas algorithms that use LER, we implement the concept of geometry-dependent LER (GLER), which was introduced by Vasilkov et al. (2017). GLER is derived from simulated TOA radiance of a Rayleigh atmosphere over a non-Lambertian surface for the specific geometry of a satellite pixel. Here "geometry-dependent" is emphasized to distinguish the GLER product (which considers the angular dependence of surface reflection) from other LER-related products or climatologies that have no dependence on sun/satellite

geometries. Our GLER approach does not require major changes to existing trace gas and cloud al-
gorithms that rely on an estimate of LER (Vasilkov et al., 2017); the main revision to the algorithms
requires replacement of the existing static LER climatologies with GLER calculated for specific
fields-of-view (FOVs) and Sun-satellite geometries. GLER can be applied to any satellite retrieval
algorithm that uses LER.

The main goal of this paper is to evaluate our GLER product over land surfaces using visible mea-
surements from the satellite-borne Ozone Monitoring Instrument (OMI). In the current version, the
GLER is based on BRDF parameters derived from MODerate-resolution Imaging Spectroradiome-
ter (MODIS) satellite observations over land; we plan to cover the ocean results in a separate paper.
We also provide additional details on the GLER methodology, including the determination of the
product components and key input model parameters. Specifically, surface BRDF models and at-
mospheric radiative transfer (RT) calculations as well as the MODIS BRDF product are introduced
in Section 2. We compare OMI-measured and simulated LER over typical geographical regions as
a function of cross-track position, season, and year in Section 3. Discussion and conclusions are
provided in the final two sections.

## 2   Data and Methods

In this section, we describe data sets and methodologies used to estimate each component of GLER.
The implementation and validation process is summarized in Figure 1. In the following, we first
briefly introduce the surface BRDF models used for GLER computation.

### 2.1   Surface BRDF models

The kernel-driven BRDF model from the MODIS BRDF/Albedo algorithm (Lucht et al., 2000) is
used in this study to describe land surface reflection anisotropy. This model is also known as the
Ross-Thick/Li-Sparse Reciprocal (RTLS). RTLS consists of a linear combination of the weighted
sum of an isotropic parameter and two kernels that characterize the scattering dependence on viewing
and illumination geometry (Roujean et al., 1992). The Ross-Thick kernel is derived from radiative
transfer models (Ross, 1981) for volume scattering within a dense vegetation canopy, and the Li-
Sparse Reciprocal kernel is based on surface scattering and geometric shadow-casting with mutual
shadowing theory (Li and Strahler, 1992).

The mathematical expression for the kernel-driven RTLS to estimate surface BRF is as follows:

$$\mathrm{BRF}(\lambda, \theta, \theta_0, \phi) = f_{\mathrm{iso}}(\lambda) + f_{\mathrm{vol}}(\lambda)k_{\mathrm{vol}}(\theta, \theta_0, \phi) + f_{\mathrm{geo}}(\lambda)k_{\mathrm{geo}}(\theta, \theta_0, \phi), \tag{1}$$

where $\theta$ is the viewing zenith angle (VZA), $\theta_0$ the solar zenith angle (SZA), and $\phi$ the relative az-
imuth angle (RAA). $k_{\mathrm{vol}}$ and $k_{\mathrm{geo}}$ are the Ross-Thick and Li-Sparse Reciprocal kernels; $f_{\mathrm{iso}}$, $f_{\mathrm{vol}}$
and $f_{\mathrm{geo}}$ are the kernel weights (also called kernel coefficients or BRDF parameters) derived every 8

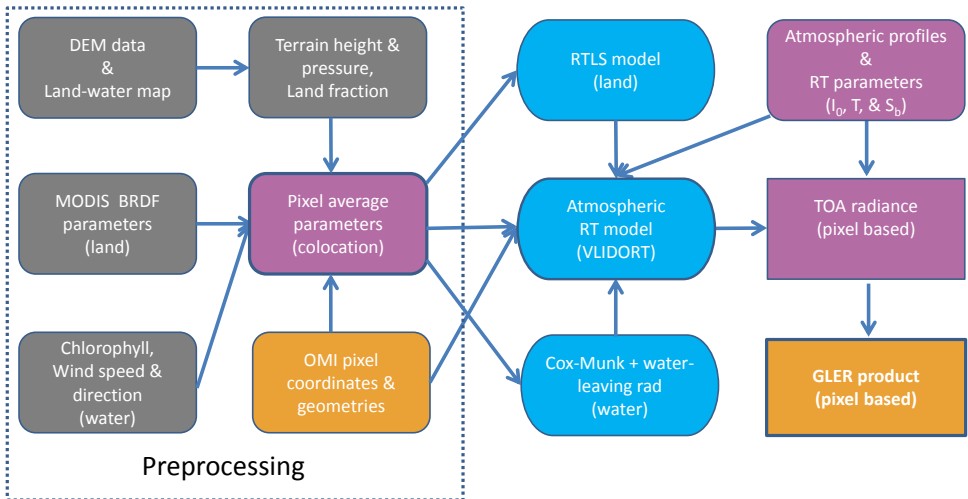

Fig. 1: Flowchart of the GLER processes. Different colors stand for different data types/sources: grey for ancillary data for both land and water, purple for preprocessed input parameters or atmospheric input parameters, gold for sensor-dependent pixel-related inputs/output, and finally, blue ovals for the physical models used. All input data are represented by the rounded rectangles and output product is shown in the box rectangle. DEM denotes digital elevation model. RTLS denotes the Ross-Thick/Li-Sparse Reciprocal functions (see Section 2.1 for details). Also see Eq. 2 for definitions of $I_0$, $T$ and $S_b$.

days by inverting the model against MODIS multi-angular observations (cloud-cleared, atmospherically corrected surface reflectances) collected for each location within a 16-day period. These kernel coefficients only depend on wavelength but not on illumination or observation angles, and have been provided globally in the MODIS gap-filled BRDF Collection 5 product MCD43GF (Schaaf et al., 5  2002, 2011).

Over water surfaces (including inland waters and oceans), light specularly reflected from a rough water surface as well as diffuse light backscattered by bulk water and transmitted through the water surface are considered. Reflection from the water surface is described by the Cox-Munk slope distribution function as implemented in Mishchenko and Travis (1997). A Case 1 water model (Morel and 10  Gentili, 1996) that has chlorophyll concentration as a single input parameter is applied to account for water-leaving radiance (i.e., light backscattered by water column into the atmosphere) including directionality of the underwater diffuse light. Chlorophyll concentration, wind speed, and wind direction are the only model input parameters.

Since the focus of this paper is on evaluating the derived GLER over land surfaces (pixel land 15  fraction $\geq$ 0.99), the brief description of the ocean models here is only for completeness. Details on water BRDF models and input datasets for wind speed and direction as well as chlorophyll concen-

tration will be provided in a separate paper for ocean GLER validation.

## 2.2 MODIS BRDF product for land surfaces

MODIS is a cross-track scanning radiometer and has 36 spectral bands ranging in wavelength from 0.4 $\mu$m to 14.4 $\mu$m. Two bands (1 and 2) have a nominal resolution of 250 m at nadir, with five

bands (3 to 7) at 500 m, and the remaining 29 bands at 1 km. MODIS views the entire Earth surface approximately daily via a two-side scan mirror that provides a swath of 2330 km cross track by 10 km along track (at nadir) each scan. The MODIS instruments are operated onboard the National Aeronautics and Space Administration (NASA) Aqua and Terra satellites, which have 16-day repeat cycles and provide measurements on a global basis every 1-2 days. MODIS data are used to study

the oceans, atmosphere and land (Justice et al., 1998). The calibration uncertainty for MODIS band 3 is within 2% (Xiong et al., 2005). The MODIS Aqua solar reflective bands including band 3 were corrected for a time-dependent drift in Collection 5 (Wu et al., 2013) but errors in MODIS Terra of up to 5% across the scan developed approximately 5 years after launch and this error was not sufficiently corrected in Collection 5 (Sun et al., 2014; Lyapustin et al., 2014).

To compute GLER, we use Collection 5 MODIS BRDF/Albedo Product MCD43 for land surfaces (Sun et al., 2017; Schaaf et al., 2002, 2011). The BRDF data in MCD43 is retrieved from surface reflectance data in the MODIS Collection 5 MOD09 product. The atmospheric correction is applied in the MOD09 product to cloud-free or partially cloud-contaminated pixels. The cloud mask also reduces thin cirrus cloud contamination (Vermote and Kotchenova, 2008). The correction removes

the effects of gas and aerosol absorption, aerosol scattering, and corrects adjacency effects caused by variation of land cover, surface and atmosphere coupling effects (Vermote et al., 2002, 2007, and 2008). The algorithm uses tables constructed with the 6SV (Second Simulation of a Satellite Signal in the Solar Spectrum Vector) radiative transfer code using key input parameters such as aerosol properties (aerosol optical thickness, size distribution, refractive indices and vertical distribution),

atmospheric pressure, ozone amount and water vapor content. These input data are described in Holben et al. (1998); Remer et al., (2005); Gao and Kaufman (2017). The atmospheric correction for MODIS band 3 used in this study has a theoretical error budget of about 0.005 reflectance units (Vermote et al., 2008). We note that the atmospheric correction neglects surface anisotropy and that Wang et al. (2010) and Franch et al. (2013) have found doing so can introduce a modest

negative bias in the corrected surface reflectance product. But despite this, Roman et al. (2013) found MODIS BRDF/Albedo products met the absolute accuracy requirement of 0.02 for spring and summer months.

Since the morning overpass (Terra) and afternoon overpass (Aqua) view the same location with different sun and viewing geometries, use of data from both satellites would double the angular sam-

ples during the 16-day repeat cycle, thus increasing the number of high quality, cloud-free observations, and reducing the uncertainty and random noise amplification of kernel coefficients retrievals

(Salomon et al., 2006; Schaaf et al., 2011). The absolute accuracy requirements for albedo for all bands in MCD43 product is 0.02 in reflectance units or 10% of surface measured values (Jin et al., 2003; Roman et al., 2013). Indeed, the majority of the extensive validation campaigns on different platforms across different landscapes and seasonal cycles have demonstrated that the MCD43 product meets this requirement. These include comparisons with ground-based or airborne measurements (e.g., Wang et al., 2004 in the Tibetan Plateau; Coddington et al., 2008 over Mexico city; Wang et al., 2012 in snow-covered tundra) as well as with space-borne data (e.g., Susaki et al., 2007 in paddy fields using Advanced Spaceborne Thermal Emission and Reflection Radiometer (ASTER) and Enhanced Thematic Mapper Plus (ETM+) data; Roman et al., 2013 with Landsat and the Cloud Absorption Radiometer (CAR) data; and Wang et al., 2014 using ETM+). However, there are a few cases where MODIS retrieved albedo are smaller than field measurements, e.g., a bias of -0.01 for the visible broadband albedo (0.3-0.7$\mu$m) over FLUXNET tower sites (Cescatti et al., 2012; Wang et al., 2010).

MCD43 provides three kernel coefficients ($f_{\text{iso}}$, $f_{\text{vol}}$, and $f_{\text{geo}}$ in Eq. 1) for 7 MODIS bands for snow-free land and permanent snow and ice cover every 8 days. Though recent improvements in the MODIS Collection 6 MCD43 BRDF data (Wang et al., 2018) may enable the use of the MCD43 data for seasonal and variable short-term snow cover in GLER product, the first version of the GLER product uses the gap filled (GF) Collection 5 product (MCD43GF) which is intended to provide BRDF parameters using the RTLS model for land surfaces free of seasonal snow and those covered by permanent snow or ice. Other snow and ice BRDF models also exist. In fact, the calibration of the OMI instrument, described in Section 2.4, is based partly on an alternate model to describe reflectance from Antarctic ice (Jaross and Warner, 2008). Because validation of snow and ice reflectances is challenging and involves different issues than those of snow-free land, we plan to carefully evaluate the GLER product over snow and ice separately in a follow-on study using various sources of BRDF information. Until that time, the GLER product over snow and ice should be considered less mature than the BRDF over snow and ice free land, whether the snow and ice are permanent (using MCD43GF), or seasonal (using OMI-derived LER) as described in Section 3.

To obtain kernel coefficients for a given OMI pixel, the collocated MCD43GF points within an OMI pixel FOV are averaged (see Appendix A1 for details). Since kernel coefficients depend on wavelength, for the present study we selected MODIS band 3, the shortest wavelength in the MCD43GF product, with a center wavelength of 470 nm (ranging from 459 to 479 nm) to represent 466 nm, which is the wavelength used in our cloud algorithm to retrieve effective cloud fraction (ECF) (Vasilkov et al., 2017). Observations at this wavelength are relatively free of atmospheric rotational-Raman scattering (RRS) and trace gas absorption.

## 2.3 Pixel land areal fraction

The areal fraction of land (or water) for each OMI pixel is a critical parameter in TOA radiance calculation for pixels mixed with land and water (see Eq. 3). However, it cannot be estimated from OMI L1b pixel surface category flags because these binary flags do not provide information on mixed pixels. Therefore, a binary land/water classification method is developed to estimate pixel land fraction from the high resolution ($30''$, same as MCD43) static land-water mask map provided with MCD43.

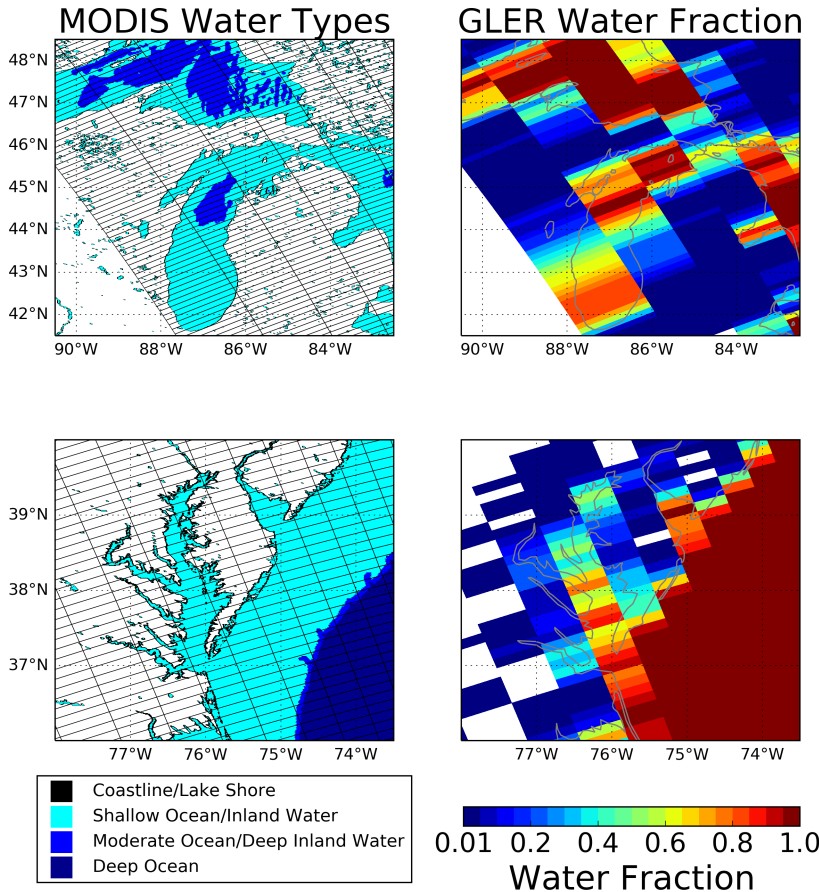

Fig. 2: Pixel water fractions (color bar) estimated in GLER (right) with MODIS $30''$ land-water map (left) with OMI pixel polygons on the top. The color legend denotes surface categories defined in the original MODIS data. Top panel: region of Lakes Superior and Michigan; Bottom panel: Chesapeake Bay.

First, we convert the eight surface categories from MCD43 into a binary land-water flag, merging all shorelines and ephemeral water at the MODIS spatial resolution into the land class and classifying

all other water sub-categories as water. The areal fraction of land (or water) for each OMI pixel is then computed from the counts of land and water points within the OMI FOV. Typical results are shown in Figure 2. Accurate estimation of pixel land fraction is also very important because BRDF models for land and water surfaces are quite different (strong backward scattering over land vs strong forward scattering over water) with different wavelength dependence. In contrast with previous studies (e.g., Zhou et al., 2010), we apply the ocean models described in Section 2.1 to coastal zones and inland waters instead of using MODIS data, because the MODIS kernel-driven BRDF model is not applicable for water surfaces.

## 2.4 OMI data and selection criteria

OMI, launched onboard the NASA Aura satellite in July 2004, is a Dutch-Finnish hyper-spectral passive imager measuring in the 270-500 nm wavelength range with two CCD detectors (UV and Vis). It was designed to provide information about trace-gases, such as $O_3$, $NO_2$, $SO_2$, HCHO, as well as absorbing aerosols. OMI has an instantaneous FOV of $0.8°$ in the flight direction (along-track) and $115°$ in the swath direction (cross-track), which yields an overall ground coverage of about 13 km by 2600 km at an altitude of 700 km. OMI measurements nominally provide daily global coverage with a 13 km × 24 km resolution in the nadir position.

OMI collection 3 data are used in this study. Specifically, we use LER retrieved from TOA radiances at 466 nm that are computed by normalizing the OMI radiances to the OMI day-1 solar irradiance spectrum measured on 21 December 2004 along with a correction for the Earth-Sun distance when calculating OMI-derived LER. The GLER product is designed to characterize the magnitude and the angular variability of the Earth's surface reflectance in a Rayleigh atmosphere, so in the context of GLER product validation, absolute radiometric response and consistency across the measurement swath are the two most critical aspects of instrument calibration to consider. For this study we ignore spectral dependence in the calibration, because our focus is strictly on the 466 nm channel. Spectral calibration will be important for validation of future versions of the GLER product that are planned to report data at several other wavelengths.

Dobber et al. (2008) estimated that the uncertainty in viewing angle dependence of OMI collection 3 sun-normalized radiances is less than 2% and the radiometric calibration uncertainty is 2%. Schenkeveld et al. (2017) evaluated long-term changes in the absolute radiometric response of the OMI instrument and estimated degradation of approximately 1-1.5% over the lifetime of the mission in the wavelength region used in this study.

Since only clear sky measurements are used for our comparison, we apply the UV aerosol index (AI) from OMAERUV product (Torres et al., 2007) to detect and screen out absorbing aerosol contaminated OMI measurements. This aerosol index is defined as the ratio of radiances measured at 354 and 388 nm compared to the ratio calculated for a pure Rayleigh-scattering atmosphere. It is sensitive to the presence of absorbing aerosols that reduce LER retrieved from OMI data. To

screen out cloud contaminated pixels in the OMI measurements, we use ECF from the oxygen dimer ($O_2 - O_2$) cloud algorithm described in Vasilkov et al. (2018). For this analysis, $|AI| < 0.5$ and ECF $< 0.04$ are used for cloud and aerosol screening (see Appendix C for more details on cloud screen selection). Data with SZA greater than 70° are not included in this analysis as the MCD43 product does not recommend the use of data beyond 70° SZA.

## 2.5 Ancillary data sets

In order to produce the pixel-level GLER product, we need first to collocate and average ancillary data that have different spatial resolutions over the OMI FOV for the physical models that we use. Table 1 summarizes the ancillary data used in terrestrial GLER production along with their spatio-temporal resolutions. This includes digital elevation model (DEM) data (ETOPO2v2) from the National Oceanic and Atmospheric Administration (NOAA). The ancillary data with higher spatial resolution than OMI are first collocated with the OMI pixel using the so-called point-in-polygon methodology described by Haines (1994) and applied by Fisher et al. (2014) in the development of a merged OMI-MODIS cloud product. Details regarding the collocation and averaging of ancillary data sets are given in Appendix A1.

Table 1: Spatial and temporal resolutions of ancillary data used for land GLER calculation

| Name | Source | Spatial | Temporal |
|------|--------|---------|----------|
| DEM | ETOPO2v2/NOAA | $2'$ | N/A |
| Land-water flag | MODIS | $30''$ | N/A |
| Land BRDF parameters | MCD43GF/MODIS | $30''$ | 8 days |

## 2.6 GLER computation

Given all necessary input parameters, TOA radiances ($I_{\text{comp}}$) are computed with the Vector Linearized Discrete Ordinate Radiative Transfer (VLIDORT) model. VLIDORT is a vector multiple scattering radiative transfer model that can simulate Stokes 4-vectors at any level in the atmosphere and for any scattering geometry with a Lambertian or non-Lambertian underlying surface (Spurr, 2006). In this study, VLIDORT computations are carried out using the pseudo-spherical correction, i.e. for both multiple and single scattering calculations, solar beam attenuation (before scattering) is treated for a spherical non-refractive atmosphere. Multiple scatter calculations are done for a plane-parallel medium. However, in the single scattering treatment, both solar-beam and line-of-sight attenuations are computed for a spherical-shell atmosphere. These "sphericity corrections" are necessary to obtain the most accurate results for geometrical configurations with large solar zenith angles, and also for wide-angle viewing scenarios. VLIDORT is executed in vector mode for our calculations, since neglect of polarization can lead to considerable errors for modeling backscattered

spectra in the UV/Vis wavelength range.

We simulate clear sky TOA radiance ($I_{\mathrm{comp}}$) over a non-Lambertian surface by coupling VLI-DORT with the MODIS kernel-driven BRDF function (Eq. 1) from the group of analytical BRDF models available in the VLIDORT BRDF supplement to account for the surface BRDF effect on

TOA radiance over land surfaces. Then GLER (or simply $R$) is defined and derived by inverting

$$I_{\mathrm{comp}}(\lambda, \theta, \theta_0, \phi, P_s, \mathrm{BRF}_s) = I_0(\lambda, \theta, \theta_0, \phi, P_s) + \frac{RT(\lambda, \theta, \theta_0, P_s)}{1 - RS_b(\lambda, P_s)}, \qquad (2)$$

where $P_s$ is the pressure at the reflecting surface, $I_0$ is the path scattering radiance by the atmosphere, calculated as the TOA radiance for a black surface, $T$ is the transmitted radiance, i.e., incident total (direct + diffuse) irradiance multiplying by transmittance from TOA to the reflecting surface along

the incoming solar beam as well as that from the surface to TOA along the satellite view direction, and $S_b$ is the diffuse flux reflectivity of the atmosphere, i.e., the fraction of upward radiance from the surface scattered back to the surface by the atmosphere (Dave, 1978). All angles are defined as in Eq. 1. The input surface BRF (i.e., $\mathrm{BRF}_s$) to VLIDORT is simulated either with Eq. 1 over land or with models described in Section 2.1 over water.

We also computed $I_0$, $T$ and $S_b$ with VLIDORT by calculating TOA radiances for three values of R, and then solving three linear equations in the form of Eq. 2 to derive the three terms. To speed up computations, however, we created lookup tables of the quantities $I_0$, $T$ and $S_b$ for different sun and viewing geometries and for a number of surface pressure levels (see Appendix B for details). Note that Eq. 2 can also be used to derive LER directly from satellite observations by simply replacing

the computed TOA radiance ($I_{\mathrm{comp}}$) with observed TOA radiance ($I_{\mathrm{obs}}$). This approach is used in Section 3, where we compute and compare OMI-derived LER to VLIDORT-simulated GLER for validation.

To make the simulated TOA radiance more realistic to a given pixel geolocation, we construct dynamical atmospheric optical property profiles using the surface (terrain) pressure, temperature and

their profiles pixel by pixel. The pressure profile is then generated following Lagrangian control volume (LCV) coordinate system starting from the surface pressure (see discussion in Appendix A2). The temperature profile is based on the Global Modeling Initiative (GMI, see Rienecker et al. (2011)) monthly climatological temperature profiles. Finally, we calculate the layer total optical thickness and single scattering albedo following Bodhaine et al. (1999) for Rayleigh cross-section calculation.

Compared with the static profiles used previously (e.g., Vasilkov et al., 2017), these dynamic atmospheric profiles better represent the actual Rayleigh atmosphere above the OMI pixel and result in a more accurate TOA radiance simulation. This dynamic profiles only apply to online calculations, whereas for this work the static profiles approach is used for look-up table (LUT) construction.

For uniform surface pixels (either 100% land or water), we calculate TOA radiance by coupling the

surface anisotropy models specified in Section 2.1 with VLIDORT. For heterogeneous surface pixels (i.e., mixed with land and water), the TOA radiance ($I_{\mathrm{comp}}$) is estimated following the independent

pixel approximation, i.e., using the area-weighted radiance from both land ($I_{\text{land}}$) and water ($I_{\text{water}}$) contributions within an OMI FOV as follows.

$$I_{\text{comp}}^{\text{TOA}} = f_L I_{\text{land}}^{\text{TOA}} + (1 - f_L) I_{\text{water}}^{\text{TOA}}, \tag{3}$$

where $f_L$ is the pixel land fraction, estimated as described in Section 2.3. Figure 3 shows examples of $I_{\text{land}}^{\text{TOA}}$, $I_{\text{water}}^{\text{TOA}}$ and $I_{\text{comp}}^{\text{TOA}}$.

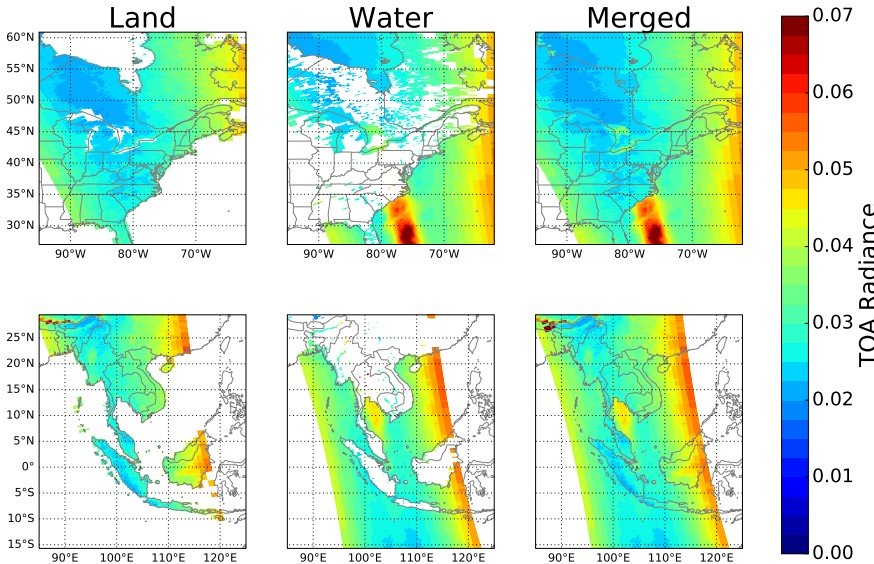

Fig. 3: Pixel-based simulated TOA radiance over land (left) when the pixel land fraction is larger than 5%, water (middle) when the pixel water fraction is larger than 5%, and the merged scene using Eq. 3 (right). Top panel: North America; Bottom panel: SE Asia.

It should be noted that aerosols are not included in the computation of the GLER. Scattering by aerosols in the atmosphere reduces the BRDF effects (Noguchi et al., 2014). Therefore, the use of the GLER may result in overestimation of the BRDF effects in the presence of aerosol and thin clouds. Our use of a retrieved ECF that implicitly accounts for the effects of non-absorbing aerosol

10 will help to alleviate this problem (Boersma et al., 2011; Lorente et al., 2018; Vasilkov et al., 2018). We plan to examine aerosol effects on GLER in a future work.

## 3 Results

First, we examine the overall performance of GLER by comparison with the OMI-derived LER, which is calculated by solving for $R$ in Eq. 2, replacing the left term $I_{\text{comp}}$ with OMI-measured

15 TOA radiance at 466 nm. We accounted for the small $O_2$-$O_2$ and $O_3$ absorption at 466 nm when

computing the quantities $I_0$, $T$ and $S_b$. When computing GLER, this was not necessary because these gases were not included in the simulation of the TOA radiances or the LUTs used to derive GLER. These $I_0$, $T$, and $S_b$ LUTs are interpolated with the sun and viewing geometry and surface pressure of a given pixel when calculating OMI-derived LER. Then we carry out an in-depth evaluation over nine typical landscapes (see Table 2 and Figure 4) covering seasonal, interannual, and cross-track variations.

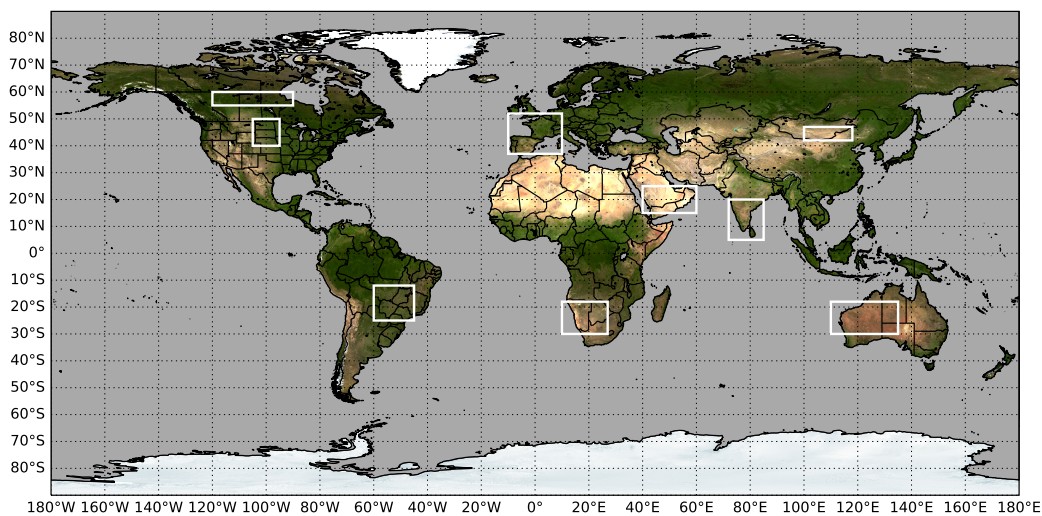

Fig. 4: Locations of selected geographical regions as specified in Table 2

.

Table 2: Selected geographical regions for analysis

| Region | Land Type | Longitude Range | Latitude Range |
|---|---|---|---|
| central Canada | Boreal Forest | 120W-90W | 55N-60N |
| central United States | Cropland/Grassland | 105W-95W | 40N-50N |
| southern Brazil | Cropland | 60W-45W | 25S-12S |
| Spain/France | Cropland/Forest | 10W-10E | 37N-52N |
| Arabian Desert | Barren | 40E-60E | 15N-25N |
| southern Africa | Open Shrubland | 10E-27E | 30S-18S |
| southern India | Crop Mosaic | 72E-85E | 5N-20N |
| Mongolia | Barren/Desert | 100E-118E | 42N-47N |
| western Australia | Open Shrubland | 110E-135E | 30S-18S |

### 3.1 Overall performance

Figure 5 shows comparisons of GLER with clear-sky OMI-derived LER at 466 nm across various geographical regions for 2006. The absolute LER varies greatly between the geographic regions; for example, forested regions exhibit LER less than 0.05 while the LER of the deserts reach nearly 0.30. Overall, the OMI-derived LER is generally higher compared with the calculated GLER, as the distribution of data fall below the 1:1 line. While this bias does seem fairly consistent from region to region, there is some small change in the magnitude.

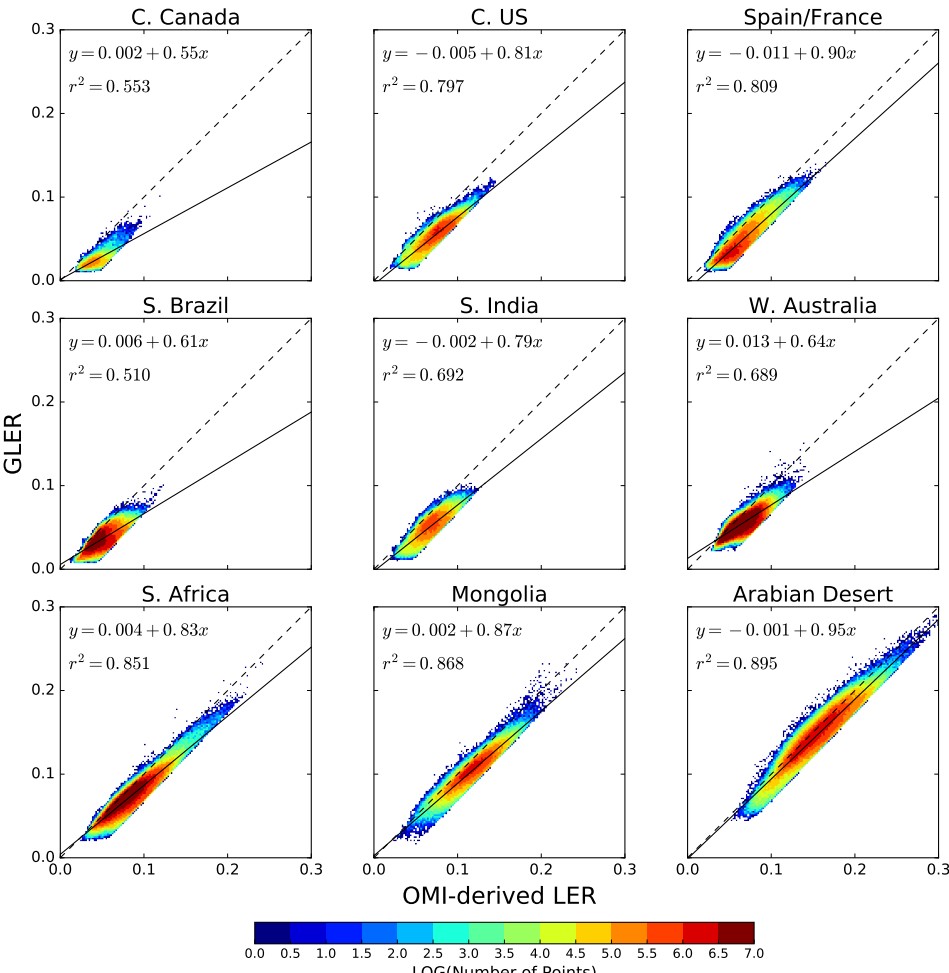

Fig. 5: Comparison of aerosol and cloud cleared OMI-derived LER at 466 nm and GLER for 2006 across various geographical regions. In the color-bar legend, N is the number of comparisons.

Despite this small bias, we note that $r^2$ is greater than 0.8 for several of the regions, with the poorest agreement in darker regions such as southern Brazil and central Canada. It is possible that in

Table 3: Seasonal Analysis of GLER

| | DJF | | MAM | | JJA | | SON | | |
|---|---|---|---|---|---|---|---|---|---|
| **Region** | **Diff** | $r^2$ | **Diff** | $r^2$ | **Diff** | $r^2$ | **Diff** | $r^2$ | **Total Count** |
| C. Canada | N/A | N/A | -0.015 | 0.61 | -0.015 | 0.40 | -0.017 | 0.56 | 20,865 |
| C. United States | -0.019 | 0.40 | -0.022 | 0.78 | -0.019 | 0.79 | -0.017 | 0.71 | 91,293 |
| S. Brazil | -0.020 | 0.43 | -0.014 | 0.34 | -0.013 | 0.50 | -0.014 | 0.54 | 197,732 |
| Spain/France | -0.020 | 0.75 | -0.017 | 0.71 | -0.014 | 0.84 | -0.019 | 0.83 | 131,959 |
| Arabian Desert | -0.009 | 0.90 | -0.007 | 0.88 | -0.007 | 0.85 | -0.009 | 0.91 | 299,860 |
| S. Africa | -0.012 | 0.86 | -0.010 | 0.87 | -0.010 | 0.81 | -0.011 | 0.82 | 379,898 |
| S. India | -0.016 | 0.66 | -0.019 | 0.67 | -0.022 | 0.72 | -0.022 | 0.60 | 78,156 |
| Mongolia | -0.016 | 0.87 | -0.012 | 0.83 | -0.014 | 0.89 | -0.013 | 0.85 | 142,770 |
| W. Australia | -0.011 | 0.61 | -0.009 | 0.67 | -0.009 | 0.75 | -0.007 | 0.66 | 161,123 |

these darker regions where the agreement worsens, the darkness of the surface maximizes the impact of residual aerosols and clouds that were not completely removed from the OMI measurements. As seen in Table 3, the agreement varies the most through the year in regions with large changes in vegetation, such as in southern Brazil where $r^2$ varies from 0.34 to 0.54. The desert regions such as the Arabian Desert show little to no change with season with $r^2$ only varying between 0.85-0.91. Overall we note that GLER is biased low when compared to OMI-derived LER by 0.01-0.02, with the largest bias over darker regions where we believe residual aerosols and clouds may play a larger role in brightening the OMI measurements. As mentioned in Appendix C, the mean of the GLER and OMI-derived LER difference may include some contribution from residual aerosol and cloud given the ECF screen used for the analysis.

### 3.2 Seasonal variations

Surface BRDF or albedo change is small on a day-to-day basis, with the exception of extreme events such as fires and floods which may not be captured with the 16 day MODIS dataset (Schaaf et al., 2011). There is, however, noticeable variability in BRDF and albedo between seasons due to land cover changes throughout the year. Since the MODIS BRDF model parameters are calculated every 8 days, they can capture the BRF and albedo changes from season to season over various land cover types. Figure 6 shows the seasonal variability of GLER, Kleipool Climatology, and OMI-derived LER for various land cover types in 2006 that have been screened for clouds and aerosols. Comparisons of OMI reflectivity data with GLER show little data across central Canada in the winter months due to the presence of seasonal snow cover, while in the southern India region, missing data occur due to persistent cloud cover during the monsoon season in the summer months.

Throughout the year, both GLER and OMI-derived LER vary as much as 0.03-0.04 at 466 nm due to changes in vegetation. The GLER follows a similar seasonal variation as compared with

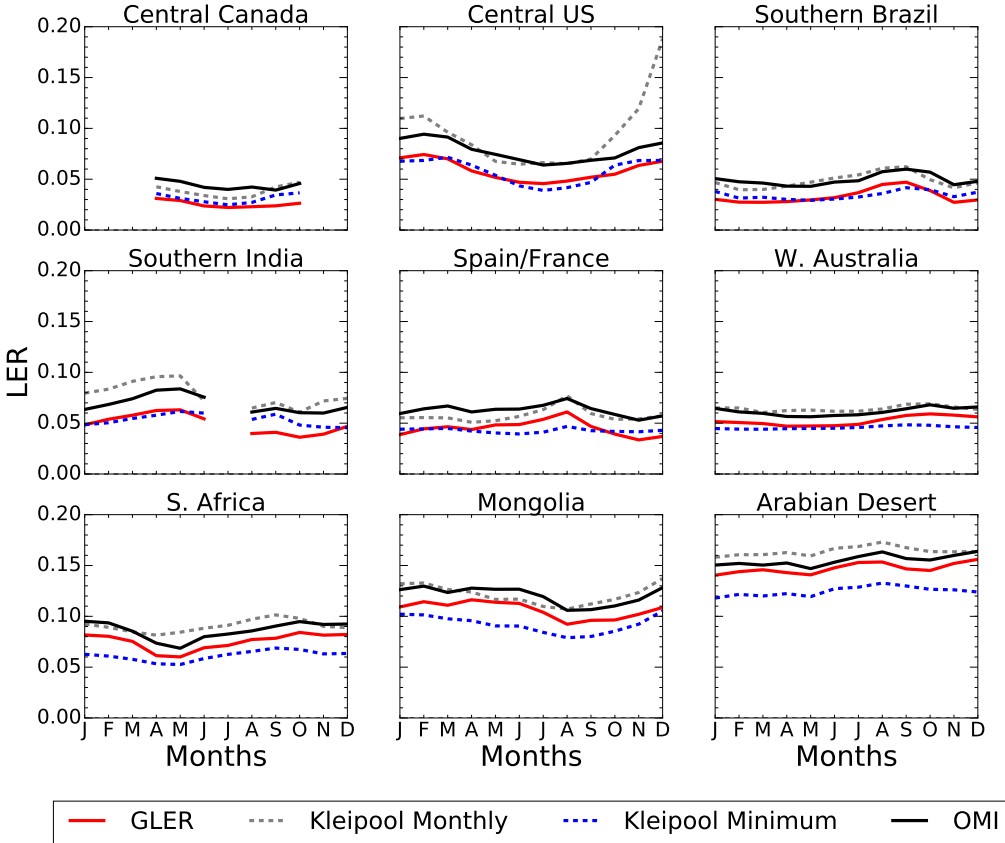

Fig. 6: Seasonal dependence in 2006 of GLER (solid red), Kleipool et al. (2008) monthly climatological LER (dashed grey), Kleipool et al. (2008) minimum climatological LER (dashed blue) and OMI-derived LER (solid black) at 466 nm.

the OMI-derived LER with an absolute difference of 0.01-0.02. We note that while the bias varies by region, there is little to no variation of the bias through the year in each individual region. The greatest agreement between GLER and the OMI-derived LER appears to be in the Arabian region possibly because the background aerosols in this high reflectance region have less impact than in other regions. The Kleipool et al. (2008) LER data exhibit the general seasonal variations seen in the OMI-derived LER but to a smaller magnitude. This is seen well in the southern Africa region where the Kleipool data show a yearly minimum in March, whereas the OMI-derived LER and GLER show the yearly minimum LER occurring closer to May. This could be due to the fact that the Kleipool data do not capture the variability that could occur year to year due to drought or anomalous rainy periods. In the winter months across the Central US, the Kleipool et al. (2008) data agree less well with the OMI-derived LER, possibly due to the presence of contamination from seasonal snow or clouds in the climatological dataset.

### 3.3 Interannual variations

When comparing results of calculated GLER against OMI-derived LER, it is important to compare data from multiple years in order to determine whether factors such as land type changes or satellite calibration drifts have an impact on the evaluation. After 2007, OMI radiances in some rows or cross-track positions are affected by an anomaly that occurred outside the instrument, producing a blockage of the intended FOV and/or scattered sunlight from outside the FOV for some rows of the CCD detectors. This is known as the OMI row anomaly, and it affects all wavelengths to some degree (see http://projects.knmi.nl/omi/research/product/rowanomaly-background.php for more information). We therefore limit the year-to-year analysis to rows 1-20 that are not impacted by the row anomaly. We also greatly minimize the impact of snow and ice misclassification and sub-pixel contamination by restricting our comparison to land surfaces below latitudes of $60°$. January and July calculated GLER is compared with OMI-derived LER for 2006 and 2015 in Figure 7. Similarly to Figure 5, the GLER values are generally biased low compared with the OMI-derived LER with a y-intercept of around -0.015 in Figure 7. There are some outliers where GLER is significantly higher than the OMI-derived LER. These data are from the Salar de Uyuni salt lake in southwest Bolivia and Lake Frome in southern Australia which only fills up during heavy rain events. These lake basins typically retain water for short periods of time and likely would not be captured in the 16 day MODIS BRDF data (Schaaf et al., 2011). The agreement of GLER and OMI-derived LER is quite similar for 2006 and 2015 with only a small increase of the slope for July 2015 as compared to July 2006.

Figure 8 shows a fairly constant bias between GLER and the OMI-derived LER, with the exception being at lower LER's where the bias decreases possibly due to the darkening of OMI LER from to shadowing from large clouds at high latitudes (Zhu et al., 2012). The differences in the July data between 2006 and 2015, though a little bit larger than those in the January data, are still within the calibration uncertainties. Given the magnitude of the difference, while it could be caused by some satellite degradation, it is possible that it could be attributed to sampling differences due to aerosol or cloud variability.

### 3.4 Cross-track dependence

Figure 9 shows LER dependence on the cross-track position across several regions with varying land types. There are two main factors that contribute to the cross-track anisotropy of LER. First and foremost is the BRDF effect. The second factor is the spatial heterogeneity of land coverage within a selected region (box) that causes a nonuniform distribution of the surface reflectivity. This effect is exaggerated for much larger pixels at the swath edges, as compared with those nearer to the nadir. We try to minimize the second effect by selecting the most uniform regions with sufficient numbers of pixels.

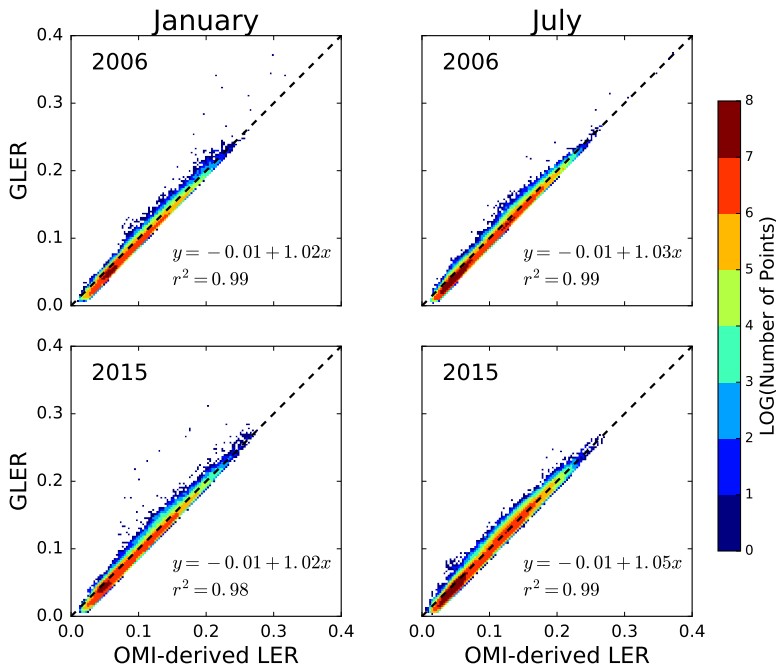

Fig. 7: Scatterplots comparing GLER with OMI-derived LER at 466 nm in January and July in 2006 and 2015, limited to rows 1-20 to exclude OMI data affected by the row anomaly. Latitudes are restricted to those below $60°$ to avoid introducing complications of snow/ice mis-classification in the comparison.

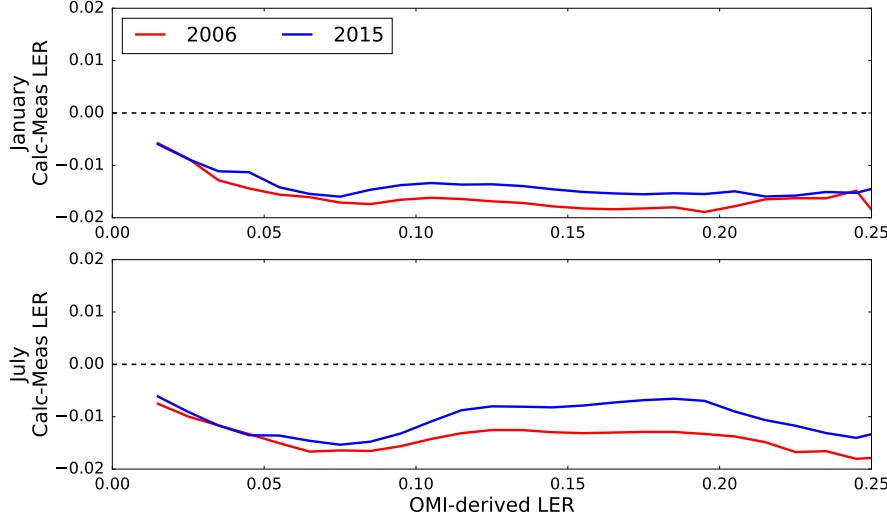

Fig. 8: Differences between calculated GLER and derived OMI LER at 466 nm plotted as a function of OMI derived LER in 2006 and 2015. Data were selected in the same way as those in Figure 7.

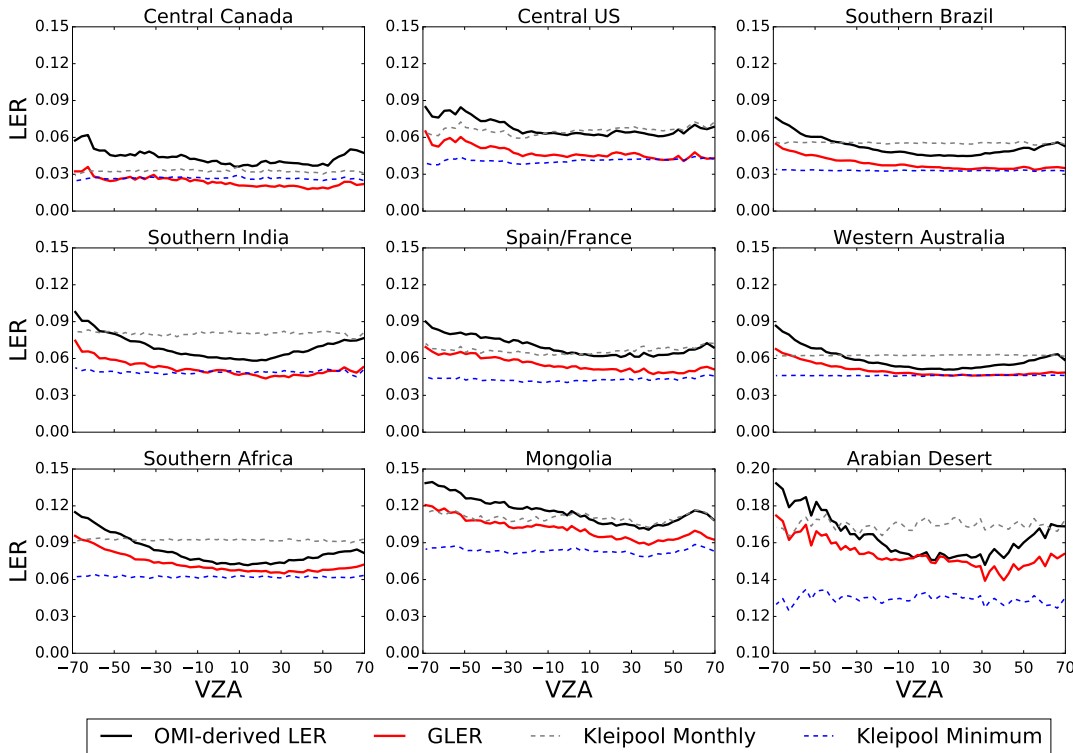

Fig. 9: Cross-track (or VZA) dependence of OMI-derived LER (black line), GLER (red line), and Kleipool et al. (2008) climatological LER (dashed line) for June-August 2006 (December-February 2006 in southern India) across various geographical regions screened for clouds and aerosols. Positive (negative) VZAs denote forward (backward) scattering directions.

However, as one can see from Figure 9, even the Kleipool et al. (2008) climatology, which has no dependence on viewing geometry, shows variations with cross-track position due to spatial non-uniformity of the surface reflectivity for some regions such as Spain/France, Mongolia, and the central US. Due to the BRDF effects, OMI-derived LER is generally larger further off nadir, in backward scattering directions. The GLER data exhibit a similar dependence, with highest values at the largest VZAs. We note that the regions that include strong absorbing dust aerosols such as the Arabian Desert and Western Australia compare well with the OMI-derived LER at nadir but there is a bias further off nadir. This could possibly be the BRDF affect from the aerosols in these regions which are not modeled with GLER since it is assumed that there is an aerosol free Rayleigh atmosphere. The darker and more forested regions such as central Canada do not exhibit the same structure in the the bias as a function of cross-track.

### 3.5 Sub-region case study

To further assess the anisotropy in GLER, we performed a small case study on a sub-region in western Australia (see Figure 10) with very homogeneous land type and elevation. Figure 11b shows that for this sub-region $f_{\text{iso}}$, which is a measure of the surface albedo, is very consistent for all rows due to the homogeneity of the surface. Figure 11a confirms the homogeneity of this region as the Kleipool et al. (2008) climatological LER is nearly constant for all cross-track positions. We note that $f_{\text{vol}} * k_{\text{vol}}$, which is a measure of the scattering of leaves and background soil/sand particulates in the scene, increases towards the edge of the swath due to increased multiple-scattering. The shadowing effect (i.e., $f_{\text{geo}} * k_{\text{geo}}$) has similar cross-track dependence in backward scattering directions, although somewhat smaller. As seen in Figure 11a, there is a similar pattern in the other regions (Figure 9). In this case study, we note that the bias becomes larger towards the edge of the swath, possibly due to the longer path length allowing for a greater impact from isolated clouds or background aerosols. Nevertheless, the overall cross-track pattern is very similar between the OMI-derived LER and the calculated GLER.

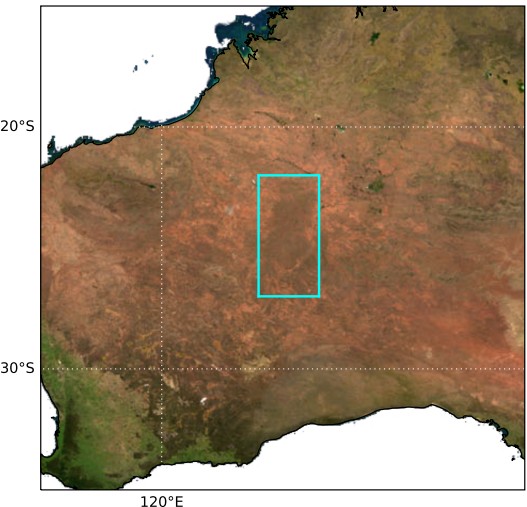

Fig. 10: Map of sub-region in western Australia with homogeneous land type used in a case study.

### 4 Discussion

Vasilkov et al. (2018) reported that values of cloud fractions derived using GLER in place of climatological LER are about 0.02 larger on average, and using GLER can significantly enhance tropospheric $NO_2$ vertical columns in polluted regions through reduction of the tropospheric air-mass factor (AMF). The results presented in Section 3 are therefore important as they demonstrate that the GLER concept as implemented with MODIS data is able to capture reliably the complex angular,

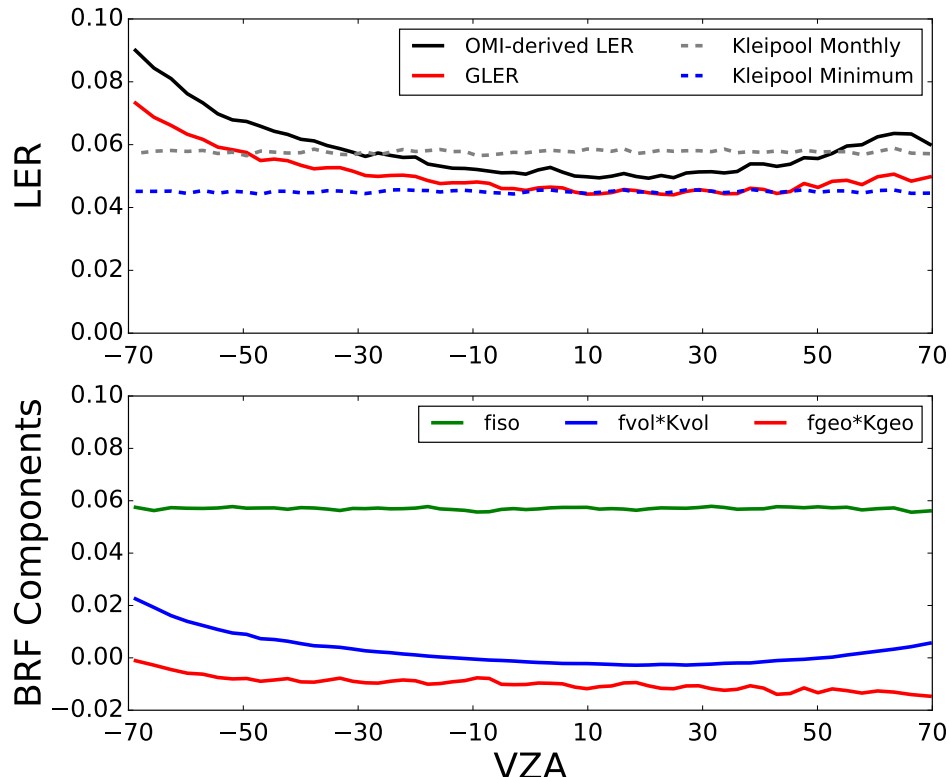

Fig. 11: Case study of cross-track (or VZA) dependence for a sub-region in western Australia in June-August 2006. 466 nm LER (top); BRF components from MODIS data (bottom). Positive (negative) VZAs denote forward (backward) scattering directions.

seasonal, and inter-annual variations in OMI reflectances over different regions on the Earth with diverse land cover types.

A significant issue related to the GLER evaluation is the presence of thin clouds and non-absorbing aerosols over land surfaces. Both effectively result in the OMI-retrieved LER being larger than
5 the calculated GLER, since neither was included in the radiative transfer simulations. Here, we excluded data with elevated cloud fractions to mitigate cloud and aerosol effects. However, the fact that OMI-derived LER is consistently biased high by 0.01-0.02 relative to GLER suggests that a certain amount of contamination is unavoidable. The effects of aerosols are partially accounted for indirectly through the current cloud algorithms that do not distinguish between clouds and non-
10 absorbing aerosol. It is therefore important that the same approach to account of surface effects, whether it be the use of climatological LER or GLER, be used for both cloud and trace-gas retrievals.

In addition to background non-absorbing aerosol and/or residual cloud contamination, it is important to consider that the GLER-LER bias may be due in part to differences in the MODIS and OMI radiance calibration. Sensitivity analysis of Eq. 2 used to compute LER and GLER shows that a 1%
15 error in TOA radiances will produce errors in LER of up to 0.003 in surface reflectivity. A bias of

0.01 between GLER and LER requires a difference in MODIS and OMI TOA radiance of at least 3% for brighter land scenes (LER $\geq$ 0.2), and differences of 10% for darker land scenes (LER $\leq$ 0.05). MODIS TOA radiances would thus have to be 3-6% low relative to OMI to explain the bias seen in GLER-LER for bright scenes, and 10-20% low for dark scenes.

Jaross and Warner (2008) compared TOA reflectances from OMI and MODIS with radiative transfer model simulations over Antarctica, accounting for the BRDF of the snow surface. By indirect comparison, OMI Collection 3 and MODIS Collection 5 agreed to within 1% at the start of the OMI mission. They estimated the uncertainty of their technique is 2%. This level of disagreement is smaller than needed to explain all of the 0.01-0.02 bias of GLER over dark scenes. We therefore con-
clude that only some of the bias can be attributed to calibration differences. Additional information about the relative calibration of OMI and MODIS is provided in Appendix D.

Relative sensor drift is also a concern in comparing the GLER product using the MODIS calibration with LER from OMI. Aqua MODIS appears to be well corrected in Collection 5 but the MCD43 product also uses data from the Terra instrument, which has degraded appreciably over the
lifetime of the mission. However, we find no evidence of time dependent change in Collection 5 MODIS BRDF data. We suspect the time-dependent and scan angle-dependent error in the Collection 5 MODIS Terra calibration data have somehow been avoided. Since OMI drift has not been fully corrected, and the MODIS drift has been removed (or avoided in the case of Terra, apparently) the slight decrease of OMI LER relative to GLER between 2006 and 2015 in Figure 8 may be due
to the 1-1.5% calibration drift in OMI radiances.

Despite these factors that introduce uncertainty into our evaluation, we conclude that the GLER product agrees remarkably well with the OMI measurements in largely clear-sky conditions. Our results suggest that GLER may be used with confidence in OMI trace gas retrievals, many of which presently utilize climatological OMI LER data. However, it should also be understood that use of
GLER calculated from aerosol-corrected MODIS BRDF data removes the effects of non-absorbing aerosols that are known to exist in the climatological LER data derived from UV/Vis sensors; this is supported by the slightly elevated OMI-derived LER we find compared with GLER.

There are other issues to be considered with the MODIS BRDF model and the Collection 5 gap-filled BRDF parameters (MCD43GF) over seasonal snow cover or permanent ice. The fact that
MCD43GF only provides snow-free land BRDF parameters usually leads to either data gaps or too small GLER values for snow-covered OMI pixels. The current temporary fix to this issue is to use OMI-derived LER but capped by a constant snow albedo of 0.6 as suggested in the KNMI's daily OMI $NO_2$ (DOMINO) product (Boersma et al., 2011; McLinden et al., 2014) based on the Near-real-time Ice and Snow Extent (NISE) flags (Nolin et al., 2005) in the OMI L1b data set. The second
issue is that the current MODIS kernel model lacks a mechanism to deal with strong forward reflection over snow/ice. Finally, since the shortest wavelength in the MODIS BRDF product MCD43GF is 466 nm, it does not cover the shorter range of OMI blue and UV wavelengths. We plan to explore

other BRDF products in the future that have more wavelengths and fewer data gaps. A good candidate is the Multi-Angle Implementation of Atmospheric Correction (MAIAC) data (Lyapustin et al., 2012). Compared to MCD43GF, MAIAC includes a shorter wavelength (412 nm) and provides pixel snow fraction that can be used for snow and ice covered regions.

We have focused here on evaluation of land GLER, because the GLER product is primarily targeted towards improvement of retrievals of trace gas pollutants such as $NO_2$ that are concentrated over land. We recognize that our evaluation in this paper excludes several important land types, such as compact and dense urban areas, land that is close to water, and a combination of the two. It can be a challenge to collect substantial amounts of data over cities, due to their relatively small size in comparison to the large regions that are the subject of this study. Particulate pollution is also common in urban regions, where non-absorbing sulfate aerosols can interfere with the derivation of LERs, thus making it difficult to validate GLERs with satellite data. These regions require further careful study using data from days when these regions are exceptionally clear. Given the importance of understanding the influence of surface reflectance on AMF calculations in highly polluted regions, we believe this work should be carried out in the future.

The validation results reported in this study apply to OMI and other sensors in similar low-Earth orbits that collect measurements with similar geometries, such as TROPOMI, which has higher spatial resolution than OMI (7 km at nadir). In theory, the smaller pixel size of TROPOMI and other future sensors should enhance the ability to validate the GLER approach by enabling more complete cloud and aerosol clearing for regions with widespread but broken clouds that were specifically avoided in the present work.

Since MCD43 product is not recommended for solar zenith angles beyond $70°$ (Schaaf et al., 2011), it may not be applicable for some geostationary (GEO) satellite observations, for which such high solar angles will certainly occur. Instead, GEO instruments such as the Geostationary Operational Environmental Satellite (GOES) imagers may be needed to provide BRDF coefficients that apply to the different range of observing conditions relevant to the planned GEO UV/Vis spectrometers.

## 5 Conclusion

The GLER product has been developed to account for surface BRDF effects on the ultraviolet and visible cloud, trace-gas, and aerosol algorithms. In this paper, we have evaluated the GLER product over land using OMI measurements for a range of land cover types. We described the atmospheric RT and surface BRDF models as well as the sources of data used in those models to produce our GLER product. Over land, the GLER product uses gap-filled Ross-Thick, Li-Sparse kernel BRDF parameters MCD43GF derived from MODIS measurements to capture the directional reflectance properties of the land surface.

We evaluated the GLER product over land by comparing it with OMI-derived LER over several typical geographical regions focusing on three aspects: seasonal variations, interannual changes, and cross-track dependence. After data are screened to remove the effects of aerosol and cloud contamination, the MODIS-based GLER show very good agreement with OMI-derived LER, with correlation coefficients larger than 0.8 for some of the selected regions. GLER also captures the seasonal variations and cross-track dependence of the OMI-derived LER. We attribute a small negative-bias of GLER data relative to OMI LER in most regions to remaining effects of non-absorbing aerosol and/or cloud contamination and to small differences in MODIS and OMI calibration. Our evaluation has demonstrated that the GLER concept can reliably and efficiently account for surface BRDF effects within UV-Vis cloud and trace-gas retrieval algorithms. In addition, GLER can be easily incorporated into the existing algorithms.

## Appendix A    Ancillary data preprocessing

### A1    Ancillary data collocation

The collocation methodology is shown schematically in Figure 12. The OMI pixel is first defined from the four ground pixel corner points provided in the OMPIXCOR data product as a 4-sided polygon. A sample space is then constructed along constant latitudinal boundaries, with the corner points tangent to the boundaries of the sample space as shown. All pixels from the MODIS BRDF/Albedo product and ancillary data sets inside the sample space are tested using the point-in-polygon method (Haines, 1994). For this application, we used the corner points for the VIS channel, corresponding to 75% of the energy in the along-track field of view. This definition assumes the pixels across the track share boundaries with their two adjacent neighbors (except for the pixels at the far edge of the swath), while the pixels along the track of the satellite overlap (reference to OMPIXCOR Readme). de Graaf et al. (2016) showed the actual shape of the OMI pixel is not exactly a rectangular polygon but rather is best represented by a super Gaussian. They also showed that the optimal overlap function between OMI and MODIS depends on the scene and the time difference between the satellites. We do not consider these factors as critical to this application because the GLER is based on MCD43GF, an 8-day gridded MODIS BRDF product from Terra and Aqua. Small errors in the pixel shape should only have a minimal impact on our results.

### A2    Pixel averaged terrain height and pressure

In order to estimate the pixel-based surface pressure, a critical input parameter to the air mass factor in the $NO_2$ algorithm as well as to total optical depth of the Rayleigh atmosphere, terrain height derived from high resolution DEM data averaged over OMI pixel FOV is required. In the GLER product, we derived pixel average terrain height from surface topographic data (ETOPO2v2), $2'$ gridded global relief data with the vertical precision of $1\,m$ from the NOAA National Centers for Envi-

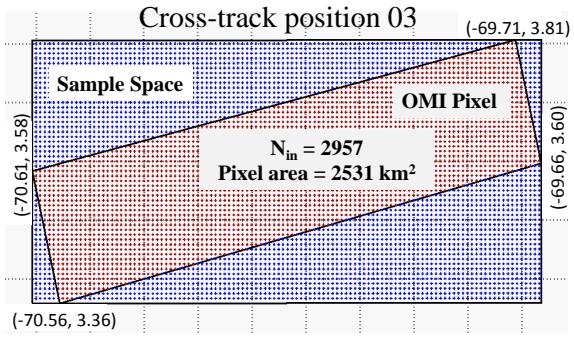

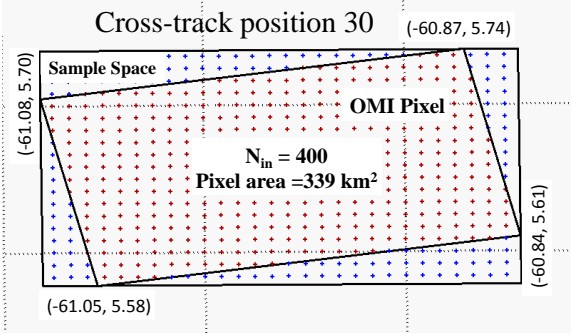

Fig. 12: An illustration of collocating the high resolution MODIS BRDF data with an OMI pixel FOV. Sample space is defined as the MODIS data space that encloses the entire OMI pixel polygon. $N_{in}$ is the number of MODIS data points within the OMI pixel (red dots). Numbers at each pixel corner indicate its geolocation (longitude, latitude). Two pixels are selected: cross-track position 03 (near the edge of swath) and cross-track position 30 (near nadir) for orbit 12399 with along-track position 1000.

ronmental Information (NCEI) Marine Geology and Geophysics (https://www.ngdc.noaa.gov/mgg/global/etopo2.html), in which positive values represent altitude above sea level while negative values represent depth below sea level. To derive the correct terrain height, we need first to determine the surface type for each ETOPO2v2 cell. This can be done by preprocessing the ETOPO2v2 data with

5 the 30″ MODIS land-water flag map described in Section 2.3. If the cell's surface type is land or inland water, we keep both positive and negative values; if it is ocean, we zero out negative values. Then we average the preprocessed ETOPO2v2 data within the OMI FOV. This approach produces a less noisy result for terrain height than the original OMI L1b terrain height which is the value at the center of the pixel (see Figure 13).

10    Given the pixel average terrain height ($z$), the terrain pressure ($P_s$) for the OMI pixel is calculated as

$$P_s = P_s(\text{GMI}) \exp\left(-\frac{z - z(\text{GMI})}{H}\right), \tag{A1}$$

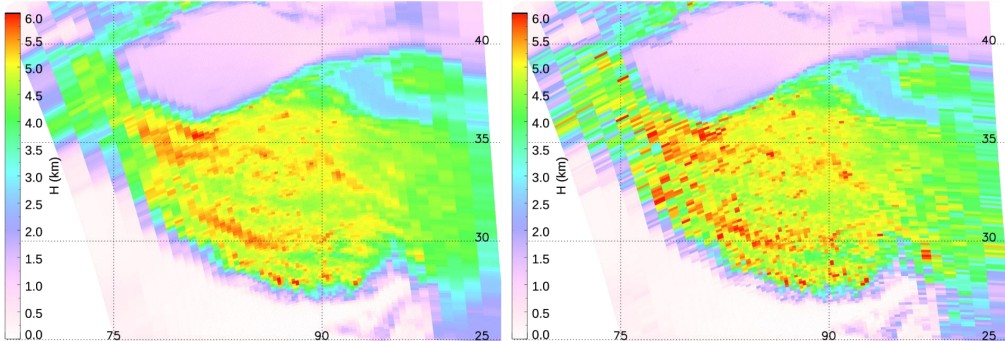

Fig. 13: Pixel average terrain height for region of the Tibetan plateau from ETOPO2v2 used in GLER (left), and OMI L1b terrain height at the center of the pixel (right).

$$H = (kT)/(Mg) \tag{A2}$$

where $P_s$(GMI) is the surface pressure monthly climatology (1° latitude by 1.25° longitude spatial resolution) taken from the Global Modeling Initiative (GMI) chemistry transport model driven by fields from the NASA GMAO Goddard Earth Observing System 5 (GEOS-5) global data as-
similation system (Rienecker et al., 2011), $z$(GMI) is the terrain height at the GMI resolution of $1° \times 1.25°$, $k$ is Boltzmann constant, $T$ is the GMI air temperature at the surface, $M$ is the mean molecular weight of air, and $g$ is the acceleration due to gravity.

## Appendix B    Look-up tables

From an operational point of view, it is impractical to process OMI and similar satellite data with on-
line radiative transfer calculations. For example, for OMI there are more than 14 years of global data, and there will be a much larger data turn-round for the recently launched TROPOspheric Monitoring Instrument (TROPOMI) and the upcoming TEMPO mission (Tropospheric Emissions: Monitoring of Pollution). Since our goal is to create a global GLER product for generic satellite missions, a look-up-table (LUT) approach is adopted to calculate variables in Eq. 2 such as $I_{\mathrm{comp}}$, $I_0$, $T$ and
$S_b$ at different surface pressure levels (see Table 4 for details). These LUTs have sufficient nodes to cover all possible OMI geometries (SZA, VZA and RAA) and model input parameters, such as three surface BRDF kernel coefficients ($f_{\mathrm{iso}}$, $f_{\mathrm{vol}}$, and $f_{\mathrm{geo}}$) for land and chlorophyll concentration, wind speed and direction for waters. The LUT approach has been validated with online VLIDORT calculations; this shows a satisfactory results of better than 0.5% relative differences between online
calculations and interpolated TOA radiances.

Table 4: LUT structures for input parameters

| Parameter | Number of nodes | Step(s) | Range |
|-----------|-----------------|---------|-------|
| Pressure | 11 | 20-110 | 411-1100 hPa |
| SZA | 45 | 2 | 0-86° |
| VZA | 41 | 2 | 0-80° |
| RAA | 48 | 2-5 | 0-180° |
| $f_{\text{iso}}$ | 25 | 0.01-0.04 | 0.01-0.999 |
| $f_{\text{vol}}$ | 16 | 0.01-0.1 | 0-0.5 |
| $f_{\text{geo}}$ | 12 | 0.005-0.02 | 0-0.1 |
| Chlorophyll | 24 | 0.003-3.0 | 0.01-10 mg/m$^3$ |
| Wind speed | 23 | 0.2-5.0 | 0.001-50 m/s |
| Wind direction | 36 | 10 | 0-360° |

**Appendix C    Cloud screen selection**

An important consideration for the method of evaluating the GLER data is properly removing cloudy scenes from the analysis. For this work, the $O_2$-$O_2$ product ECF was used for the removal of cloudy OMI scenes because this ECF will be used in $NO_2$ retrievals that the GLER product aims to improve.

Since the $O_2$-$O_2$ ECF depends on the GLER and OMI TOA Radiances, care was taken into analyzing the distribution of the GLER and OMI-derived LER difference for various cloud fraction cutoffs. Figure 14 shows the distribution of the difference between GLER and OMI-derived LER across various geographical regions for five different possible ECF cutoffs. The mode of the distribution likely represents the majority of cloud free scenes and can be thought of as a representation of the

bias between the calculated GLER and measured OMI-derived LER. We note that on the right side of the distribution there is a small tail where OMI-derived LER is less than GLER. This could be caused either by the uncertainty of the MODIS measurements or absorbing aerosols that are not being completely removed with the AI screen. To capture both the mode of the distribution as well as possible noise in the measurements which could be within the right tail of the distribution, the

ECF cutoff of 4% is used for the evaluation. We note that this cutoff may leave some residual clouds in regions such as western Australia where the left tail is larger than the right tail, but have decided to use a consistent cloud screen for all regions and note that the extra number of possible cloud contaminated data in the left tail are much less than the number of data within the mode of the distribution.

While these histograms suggest that the mode of the GLER - OMI-derived LER difference is likely most representative of the cloud free OMI scenes, in this work we present the mean of the difference. In the evaluation we examine regions such as that include month with extreme cloudiness or constant snow cover. For these months, since the number of available data are limited for evaluation, the

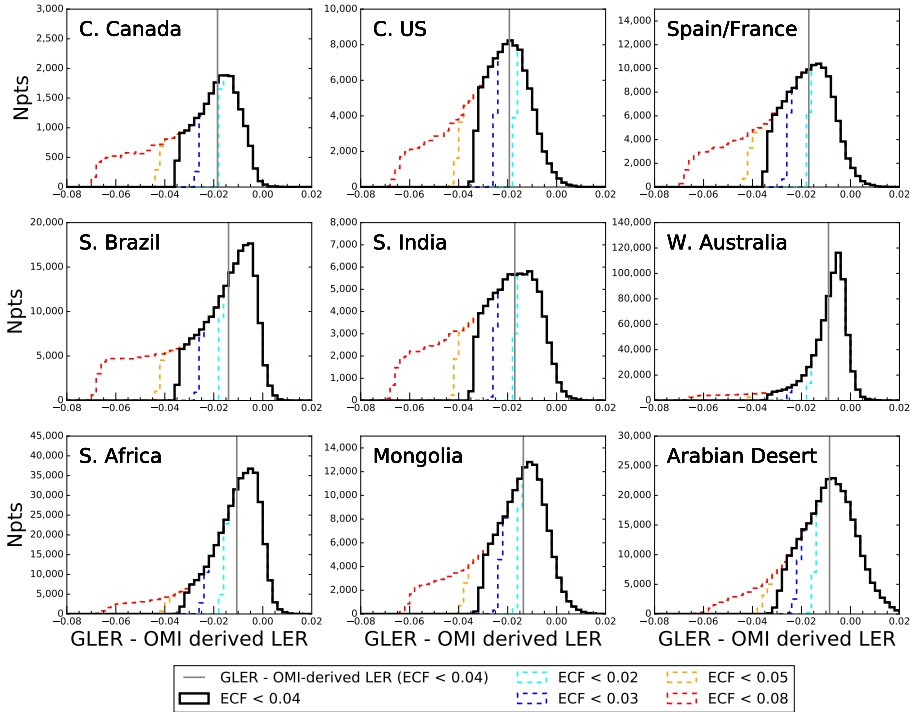

Fig. 14: Histograms showing the distribution of the difference between GLER and OMI-derived LER for various $O_2$-$O_2$ ECF screenings across different geographical regions in 2006. While no cloud screening was performed, aerosols were removed and OMI scenes with land fraction $< 99\%$ were not included. The vertical grey line represents the mean of the difference between GLER and OMI-derived LER for the various regions. The dark black line representing ECF $< 0.04$ was the cloud screen implemented in the evaluation of GLER for this paper.

distribution of the data becomes quite flat making the mode difficult to determine. As shown in Figure 14 the mean of the difference is nearly identical to the mode in regions such as the Arabian Desert and Central United States, while in other regions such as southern Brazil and southern Africa the mean of the difference is lower than the mode difference by nearly 0.01. This possibly suggests

5  that the mean of the difference is more influenced by the residual cloud and aerosol than the mode of the difference. For this reason, it is possible that the bias between the calculated and measured LER is slightly smaller than the mean difference presented in this work.

## Appendix D    Relative calibration of OMI and MODIS

Jaross and Warner (2008) compared 2004-2005 radiances from OMI and MODIS to TOA radiance

10  predicted using a radiative transfer model over Antarctica. We use these results to indirectly compare the calibration of OMI and MODIS radiances. Figure 9a of Jaross and Warner, reproduced here as Figure 15, shows that MODIS band 3 reflectances near 470 nm in Collection 4 data were around

1% high relative to the model at nadir, and OMI Collection 2 L1B data were approximately 2.5%
lower than the model for similar viewing conditions. Based on this result, the OMI calibration
team applied a +2.5% time-independent, wavelength-independent calibration adjustment to OMI
Collection 3 radiances to bring the L1B into agreement with the model (Dobber et al., 2008; Jaross
and Warner, 2008). The MODIS radiance calibration was unchanged between Collections 4 and 5
for the period Jaross and Warner examined, therefore the MODIS Collection 5 radiances are higher
than OMI Collection 3 radiances by approximately 1%. This is within the 2% uncertainty estimated
for the model. The agreement is also within the theoretical combined uncertainty calculated from

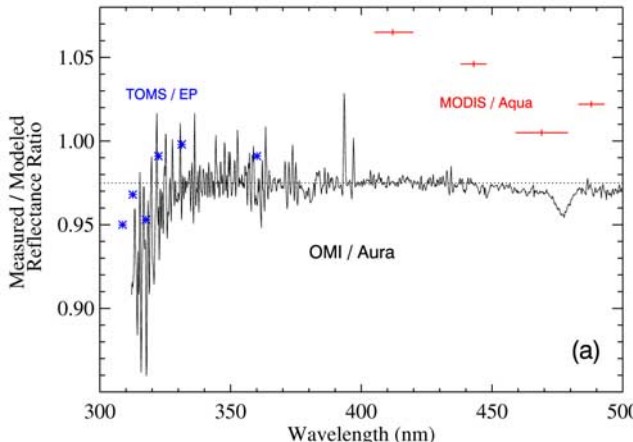

Fig. 15: The mean ratios of TOA satellite reflectances to a model used over Antarctica for valida-
tion during the 2004 solstice; OMI (solid line), MODIS (red bars), and TOMS/Earth Probe (aster-
isks). Data shown are for the interval $62° \leq \theta_0 \leq 68°$ ($47° \leq \theta_0 \leq 52°$ for TOMS/Earth Probe).
Uncertainties on MODIS bands other than band 3 (470 nm) are estimated to be several percent.
Reproduced from Figure 9a of Jaross and Warner (2008).

the uncertainties reported for each instrument independently in the literature. The error in OMI
radiometric calibration at nadir is 2%, uncertainty in swath dependence is also within 2% (Dobber
et al., 2008), so we estimate the combined calibration uncertainty of OMI is 2-3%. The MODIS
total uncertainty is 2% (Xiong et al., 2005), therefore the theoretical combined uncertainty in the
difference between OMI and MODIS is 3-4%. In order to explain the bias in GLER - OMI LER
of 0.01 to 0.02, the MODIS radiances would have to be biased 3-6% low relative to OMI, so it is
unlikely that calibration difference is the main cause of the bias in our GLER comparisons. The
bias is most likely due to a combination of the relative calibration differences and the presence of
residual cloud and aerosol contamination that increase the measured OMI radiances relative to those
we simulate with GLER.

Time dependent degradation of the instruments is also a factor to consider when comparing the
relative instrument calibration. Jaross and Warner performed their analysis with the first few years of

overlap in the OMI and MODIS data and did not examine long-term instrument drift. Schenkeveld et al. (2017) estimated that the long-term degradation of OMI reflectances at 466 nm is 1-1.5% from 2004 to present. This drift has not been corrected in the Collection 3 L1B radiance or the GLER products. The MODIS Aqua solar reflective bands including band 3 were corrected for time-

dependent drift in Collection 5 (Wu et al., 2013), but errors in MODIS Terra due to anomalous degradation of up to 5% across the scan appeared around 2007, and this error was not sufficiently corrected in Collection 5 (Lyapustin et al., 2014). We see no evidence of the impact of such a drift on the GLER product or the BRDF itself, so we suspect that poor quality MODIS Terra data were excluded when the MCD43GF product was generated.

*Data availability.* GLER is available at https://aura.gesdisc.eosdis.nasa.gov/data/Aura_OMI_Level2/. The MODIS gap-filled BRDF Collection 5 product MCD43GF used for calculation of GLER in this paper is available at ftp://rsftp.eeos.umb.edu/data02/Gapfilled/ (last access: 11 March 2019). The OMI Level 1 data used for calculations of GLER are available at https://aura.gesdisc.eosdis.nasa.gov/data/Aura_OMI_Level1/ (last access: 11 April 2019). The OMI Level 2 Collection 3 data that in-

clude $NO_2$ and OMI pixel corner products are available at https://aura.gesdisc.eosdis.nasa.gov/data/Aura_OMI_Level2/ (last access: 11 April 2019). OMI $O_2-O_2$ Cloud product can be provided upon request of the co-authors.

*Author contributions.* WQ led the paper and was the primary developer of the GLER algorithm. WQ, ZF, and DH wrote the paper. ZF and WQ performed the GLER analysis. ZF and DH designed

the GLER analysis. AV and NK contributed to the GLER analysis and preparation of the paper. JJ provided guidance throughout the development of the manuscript. BF contributed tools and expertise for MODIS-OMI collocation. RS developed the VLIDORT code used for the BRDF and radiance computations.

*Competing interests.* The authors declare that they have no conflict of interest.

*Acknowledgements.* Funding for this work was provided by NASA through Aura core team funding as well as the Aura project and Aura Science Team and Atmospheric Composition Modeling and Analysis Program managed by Kenneth Jucks and Barry Lefer. We acknowledge Crystal Schaaf for providing gap-filled MODIS MCD43GF BRDF data, the MODIS data processing team, as well as the OMI calibration and data processing teams at KNMI and NASA. We gratefully acknowledge helpful discussions with Alexei Lyapustin, Crystal

Schaaf, Glen Jaross, members of the MODIS Characterization Support Team, and Sriharsha Madhavan.

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
