# Peer review of "A geometry-dependent surface Lambertian-equivalent reflectivity product at 466 nm for UV/Vis retrievals: Part I. Evaluation over land surfaces using measurements from OMI"

_Atmospheric Measurement Techniques, 2018_

## Referee Comment (RC1) · Anonymous Referee #1 · 9 Feb 2019

The manuscript by Qin et al. presents the generation and test of a geometry-dependent Lambertian-equivalent reflectivity (albedo) product. This 'GLER' product is derived from MODIS products that characterize the surface reflectance anisotropy at high spatial resolution. The purpose is of the GLER-product is to replace the conventional approaches making use of a viewing-geometry independent LER, starting with OMI. Now that the spatial resolution of sensors such as OMI, TROPOMI and upcoming geostationary instruments is becoming better than 10 km, the effects of surface reflectance anisotropy are increasing in relevance and (also) need to be accounted for in satellite

retrievals from TROPOMI and TEMPO, GEMS, and Sentinel-4.

The paper is generally well written, but a bit long (and are the 40 scatterplots really necessary?). The thorough introduction to the topic is valuable, as well as the good and useful references. The test of the GLER product against the OMI LER and the standard viewing-geometry Kleipool et al. [2008] climatology over different regions and years is solid. I very much appreciated that the authors reiterated the point that a consistent approach for surface reflectivity should be taken for both cloud and trace gas retrievals, as the latter depend on consistently derived cloud information. This is not always done. There are OMI NO2 products around that use BRDF-parameters for calculating the clear-sky AMFs and then a LER for the clouds, and such inconsistencies lead to avoidable systematic errors in the data product.

I have a two main concerns that I feel should be addressed.

**1. Darker MODIS than OMI scenes**

The authors make a deliberate choice to generate a GLER product based on measure-ments from another instrument (MODIS) than the product will be used for (OMI). This is understandable since kernel coefficients describing surface reflectance anisotropy are not available from the OMI sensor itself. The drawback however is that the GLER product is based on a set of very different viewing conditions, geometries, assumptions on the state of the atmosphere, and instrument specifics. All these inconsistencies can make the GLER product potentially less suitable for application on OMI retrievals. The authors are surely aware of this, and discuss some of these differences (such as the higher probability that the larger OMI scenes have been influenced by residual clouds and aerosols), but provide too little information on others. Since the MODIS-based GLER is proposed as the preferred ancillary dataset for future NO2 and O2-O2 cloud retrievals, we need to learn more about the (hopefully good) representativeness of the MODIS-based data for the OMI scenes. The MODIS atmosphere-corrected BRDF co-efficients are crucial in this sense, and we need to obtain confidence in the GLER product. Yet the atmospheric correction for the MCD43 product is hardly discussed (only briefly on page 5). While some relevant papers are cited, it is unclear how the MCD43 product accounted for the presence of clouds, aerosols, and atmospheric pressure. The authors should

*a. explain how the atmospheric correction was done, and*

*b. how the correction and/or the MODIS data screening may have led to an ensemble of (MODIS) scenes that is generally 'darker' than the OMI scenes.*

Without such clarifications, it remains misty whether allegedly "small calibration differences" between MODIS and OMI could explain the differences between the GLER and LER, and whether the MODIS-based GLER product is actually so suitable as claimed by the authors.

**2. Water model**

For inland waters, ocean models are used, but the manuscript remains vague on how the water reflectance anisotropy is accounted for in the approach. The authors should provide a mathematical description of how the GLER is computed for ocean scenes. The Appendix A doesn't cut it, as only ancillary data used to calculate the surface reflectance anisotropy rather than the actual formulas are given.

**Specific comments:**

P3, L2-4: the point that surface anisotropy effects are more relevant in NIR than in the VIS was prominently made in Lorente et al. [AMT, 2018], and it would be appropriate to cite that paper here.

P5, L22-25: it is not clear why the authors include the phrase about the use of both morning and afternoon MODIS sensors, since this is not 'aan de orde' in the manuscript.

P5, L26-34: can the authors be more quantitative here and state the quantitative find-

ings from the albedo validation exercises? Any indications for the MODIS albedo being biased low or high? What were the "accuracy requirements" exactly?

P7, suggest to move Figure 2 to the Supplement. I think the readers can trust the experienced NASA-team to do a proper job in re-gridding, and there is no new science in here.

P8, L6: please clarify what "day-1 solar irradiance spectrum" refers to. Is it the irradiance spectrum measured on 1 October 2004?

P9, Figure 3 also appears redundant. I don't see why these (quite common) re-gridding approaches should be discussed in detail. The figure looks to me as a mere illustration of the approach described in Haines et al. [1994], so I'm afraid nothing's new here.

P10, L10-12: it is unclear how application of a pseudo-spherical geometry calculation can lead to a "sphericity correction for both incoming and outgoing viewing directions". Please discuss this in more detail. How does the supposed spherical correction relate to the pseudo-spherical correction only?

P10, L19: in Eq. (2) and from the text below it is not immediately clear that Icomp refers to the VLIDORT-simulated TOA radiance levels based on simulations with a pure Rayleigh atmosphere and the capacity of the model to account for the surface BRF. Then, R can only be found if the model can simulate I0, T, and Sb, something that VLIDORT surely can, but is not becoming clear from the text.

P11, L26-27: it would be appropriate to cite papers here that made the point that cloud fraction retrievals actually provide 'effective cloud' fraction information that accounts for aerosol effects, e.g. Boersma et al. [2011], and for higher scattering in the forward direction of cloud particles, e.g. Lorente et al. [2018].

P11, L31: suggest to clarify that "this equation" refers to Eq. (2).

P13, L1-5: can you provide a quantitative statement on how much OMI LER is typically higher than GLER? From the intercepts one would say the difference is 0.01, but possibly the mean or median difference is a more meaningful metric. I would also encourage if the authors could report whether there is a pattern in how the LER-GLER differences change between different regions/surface types.

P15, L12-14: the hypothesis that localized floodplains darken after rain resulting in a signal detected by OMI LER (daily data), but not by GLER (MODIS-based 8-day data) needs to be substantiated. It sounds possible, but there is no basis for this statement from a result shown.

P16, L5-7: it is possible that OMI data is indeed affected by residual clouds or aerosols leading to higher reflectances. But it is also possible that the MODIS-based data have been overcorrected for atmospheric effects. As long as no evidence is presented to obtain confidence in the validity of the atmospheric correction (and data screening) applied to the MDC43 suite, we cannot know if it's one or the other, see my main concern.

P19, L8-10: please explain how using the GLER reduces the tropospheric AMF. Is it via the increased cloud fractions (more screening), or the lower clear-sky AMFs because of the darker surface, or both?

P19, L20: please clarify how differences in calibration between MODIS and OMI could explain the bias of 0.01 between OMI LER and GLER. Is there any reason to believe that OMI is calibrated such that it detects too low, or MODIS too high reflectances? Have level-1 data been compared in the first place?

---

## Referee Comment (RC2) · Anonymous Referee #2 · 18 Feb 2019

This paper introduces an interesting approach in which MODIS BRDF information is used to calculate a geometry-dependent surface LER (GLER) for each of the measurements collected by the OMI instrument. Users of OMI data can very easily include surface BRDF information in their retrieval algorithms by using the pre-calculated GLER product.

The topic is fitting for AMT. The paper is interesting to read and well written. I think this paper deserves publication.

I do have some comments and questions:

(1)

The paper provides some explanations for the differences that are found between the MODIS-based GLER and the OMI-derived LER. These explanations are neglect of aerosol in the simulations, the possibility of cloud contamination in the OMI observations, and calibration differences between the MODIS and OMI instruments.

What is actually known about the calibration of MODIS compared to that of OMI? Haven't there been any studies comparing the two instruments? What are the calibration differences needed to explain the systematic differences between the MODIS-based GLER and the OMI-derived LER?

For the GLER no OMI measurement data are used. Only the OMI ground pixel extent and geolocation are used and MODIS BRDF data are then used to calculate the GLER for that OMI pixel. So, the MODIS-derived GLER, which is to be used for OMI retrievals, inherited the calibration differences that exist between MODIS and OMI.

It might be good to mention (more explicitly) in the paper that using the GLER for OMI-based retrievals can result in the introduction of calibration inconsistencies in these retrievals.

(2)

What are, to your knowledge, the expected differences between MODIS-derived GLER and the MODIS BRDF? In other words, what are the errors the user would introduce if he/she would use the MODIS-derived GLER as if it would be a BRDF?

Is the 0.8% mentioned in section 2.1, page 5, lines 18-19 a representative percentage for this? Is this 0.8% valid for 466 nm?

Does the term "green band" in line 20 refer to the 555-nm MODIS band (band 4)?

(3)

The paper introduces GLER calculated at 466 nm. For this, MODIS BRDF information is used from MODIS band 3. However, MODIS band 3 is centered around 470 nm, and not around 466 nm.

On page 7, lines 6-7, an explanation for the 4-nm difference is given which suggests that the 470-nm MODIS BRDF is just used "as is" at 466 nm.

If this is the case, wouldn't it be better to say that the GLER is representative for 470 nm (and not for 466 nm, even though the GLER retrieval is performed at this wavelength)?

What are the technical reasons for doing the GLER retrieval at 466 nm and not at 470 nm? Is it because of absorption by trace gases at 470 nm? If so, which trace gases?

(4)

section 1, page 3, line 9:

Here the paper mentions a few papers on LER databases. Could you also mention the databases/instruments behind these references?

Please also add the (more recent) reference to Tilstra et al. (2017), about LER retrieval from the GOME-2 instrument. https://doi.org/10.1002/2016JD025940

(5)

section 2.3/Fig. 3:

When explaining how the point-in-polygon method is used, please mention briefly that the "real" OMI pixel is not rectangular, and that this fact alone already can lead to (small) differences between the (MODIS-derived) GLER and the OMI-derived LER.

Perhaps you could refer to the paper by De Graaf et al. (2016) about the size and shape of the OMI pixels. https://doi.org/10.5194/amt-9-3607-2016

In this paper, OMI and MODIS (band 3) reflectances are compared to each other using different sizes and shapes for the OMI point-spread function.

(6)

section 3.2, page 13, line 16:

The paper mentions here that surface BRDF does not change on a day-to-day basis. But this can happen in certain cases, as explained in section 3.3, page 15, lines 12-14.

Perhaps you could change the sentence to "While surface BRDF in general does not ..." (or something similar)?

(7)

In the paper the GLER and the OMI-derived LER are also compared to the Kleipool climatology. Which field is taken from the Kleipool climatology? Is it the field "MonthlyMinimumSurfaceReflectance" or is it the field "MonthlySurfaceReflectance"?

If it is the "MonthlySurfaceReflectance" field, then that would probably explain part of the higher values of the Kleipool climatology compared to the OMI-derived LER for at least some of the land cover types in Figure 9. In fact, I think that in these analyses it would be better to use the traditional "MonthlyMinimumSurfaceReflectance" field.

In any case, it would be good to mention in the paper which of the two fields was used in the analyses.

(8)

small typo in a reference, page 27, line 20:

Haines, E.,: –> Haines, E.:

───────────────────────

---

## Author Comment (AC1) · 7 May 2019

Dear Editor,

We thank the reviewer for the evaluation of our paper and useful comments that helped to improve the manuscript. Below are our responses to each comment. Reviewer's comments are in blue, the responses are in black; the text added to the manuscript is in red.

On behalf of the authors,

Wenhan Qin

**Responses to comments from Referee #1:**

1. Darker MODIS than OMI scenes

The authors make a deliberate choice to generate a GLER product based on measurements from another instrument (MODIS) than the product will be used for (OMI). This is understandable since kernel coefficients describing surface reflectance anisotropy are not available from the OMI sensor itself. The drawback however is that the GLER product is based on a set of very different viewing conditions, geometries, assumptions on the state of the atmosphere, and instrument specifics. All these inconsistencies can make the GLER product potentially less suitable for application on OMI retrievals.

The BRDF kernel coefficients retrieved from the MODIS data are theoretically independent of the viewing geometry because they are derived by fitting the kernel-driven BRDF model (see Eq.1) against atmospherically corrected angular observations collected during a 16-day period from both Terra and Aqua. This means one can reconstruct the entire surface BRF and compute the directional reflectance at any combination of solar and view angles desired. The BRDF parameterization is therefore a transfer function from the original MODIS source measurements to the target measurement geometry of OMI. We revised Section 2.1 to make this clearer (see marked up copy). The reconstructed BRF may have error if the MODIS geometries that the BRDF kernel coefficients are derived from do not adequately span the range of the OMI geometries. However, this is not the case for OMI and the MODIS instruments. Aura and Aqua fly in similar orbits with similar viewing geometries in the NASA "A-train" satellite constellation crossing the equator at around 1:30 pm local time. Use of Terra MODIS data for the BRDF

product increases the amount of cloud-free data that can be used to retrieve BRDF information. The MODIS instruments on Terra and Aqua observe the Earth three hours apart, at 10:30 am and 1:30 pm.

We added the following in the Section 2.1.

fiso, fvol and fgeo are the kernel weights (also called kernel coefficients or BRDF parameters) derived every 8 days by inverting the model against MODIS multi-angular observations (cloud-cleared, atmospherically corrected surface reflectances) collected for each location within a 16-day period. These kernel coefficients only depend on wavelength but not on illumination or observation angles, and have been provided globally in the MODIS gap-filled BRDF Collection 5 product MCD43GF (Schaaf et al., 2002, 2011).

The authors are surely aware of this, and discuss some of these differences (such as the higher probability that the larger OMI scenes have been influenced by residual clouds and aerosols), but provide too little information on others. Since the MODIS-based GLER is proposed as the preferred ancillary dataset for future NO2 and O2-O2 cloud retrievals, we need to learn more about the (hopefully good) representativeness of the MODIS-based data for the OMI scenes. The MODIS atmosphere-corrected BRDF coefficients are crucial in this sense, and we need to obtain confidence in the GLER product. Yet the atmospheric correction for the MCD43 product is hardly discussed (only briefly on page 5). While some relevant papers are cited, it is unclear how the MCD43 product accounted for the presence of clouds, aerosols, and atmospheric pressure.

We agree with the reviewer that the representativeness of the MODIS-based data that we propose to use in OMI trace gas and cloud retrievals is an important concern, and therefore a discussion of the atmospheric correction is warranted.

1a. explain how the atmospheric correction was done

We have added this description of the atmospheric correction to section 2.2

The BRDF data in MCD43 is retrieved from surface reflectance data in the MODIS Collection 5 MOD09 product. The atmospheric correction is applied in the MOD09 product to cloud-free or partially cloud-contaminated pixels. The cloud mask also reduces thin cirrus cloud contamination (Vermote and Kotchenova, 2008). The correction removes the effects of gas and aerosol absorption, aerosol scattering, and corrects adjacency effects caused by variation of land cover, surface and

atmosphere coupling effects (Vermote et al., 2002, 2007, and 2008). The algorithm uses tables constructed with the 6SV (Second Simulation of a Satellite Signal in the Solar Spectrum Vector) radiative transfer code using key input parameters such as aerosol properties (aerosol optical thickness, size distribution, refractive indices and vertical distribution), atmospheric pressure, ozone amount and water vapor content. Holben et al. (1998), Remer et al. (2005), and Gao and Kaufman (2003) describe these input data. The atmospheric correction for MODIS band 3 used in this study has a theoretical error budget of about 0.005 reflectance units (Vermote et al., 2008). We note that the atmospheric correction neglects surface anisotropy and that Wang et al. (2010) and Franch et al. (2013) have found doing so can introduce a modest negative bias in the corrected surface reflectance product, but despite this, Roman et al. (2013) found MODIS BRDF/Albedo products met the absolute accuracy requirement of 0.02 for spring and summer months.

1b. how the correction and/or the MODIS data screening may have led to an ensemble of (MODIS) scenes that is generally 'darker' than the OMI scenes

It is possible that errors in the correction of MOD09 data could in part be responsible for the lower relative values the GLER derived from MODIS. But the combined impact of the uncertainties and systematic errors does not appear large enough to explain all of the difference between the MODIS-derived GLER and OMI LER. Residual cloud and background aerosol contamination in the OMI measurements that even in small amounts can increase retrieved LER are likely to contribute equally or more to the 'darker' character of the MODIS scenes.  The methods of screening the OMI data we used in this study were fairly simple cut-offs of cloud fraction and UV aerosol index and unsurprisingly we found that our comparisons were sensitive to the choice of these thresholds.  This study has highlighted the importance of taking into account residual aerosol and cloud effects in efforts to analyze "clear-sky" scenes.  As part of future research, we will develop better methods to do this, or perhaps correct the OMI data for scene brightening effects.

2.  Water model

For inland waters, ocean models are used, but the manuscript remains vague on how the water reflectance anisotropy is accounted for in the approach. The authors should provide a mathematical description of how the GLER is computed for ocean scenes. The Appendix A doesn't cut it, as only ancillary data used to

calculate the surface reflectance anisotropy rather than the actual formulas are given.

We had not provided a detailed mathematical description of the treatment of inland waters with the ocean model because our focus in this paper is solely on the evaluation of land GLER only. We will include the details of the water model in a following paper evaluating out GLER product for oceans, inland waters, and scenes with a mixture of water and land. We have revised section 2, adding subsection 2.1 to briefly describe the water BRDF model for the benefit of readers, and provide the reference to Vasilkov et al. (2017) which has more information. We also removed the former Appendix A to avoid confusion and keep the focus of this paper on the evaluation over land.

Specific comments

P3, L2-4: the point that surface anisotropy effects are more relevant in NIR than in the VIS was prominently made in Lorente et al. [AMT, 2018], and it would be appropriate to cite that paper here.

We agree about the relative influence of surface anisotropy in different spectral regions. In the Introduction (first paragraph of page 3), we added a brief discussion of this topic and reference Lorente et al. (2018). We do believe however that the effects of surface reflectance anisotropy on cloud and trace gas retrievals are non-negligible in the visible region since relatively small changes in surface reflectivity can affect cloud fraction and trace gas AMF.

We modified the first paragraph of page 3 as follows.

As a result, the surface anisotropy's impact on TOA radiance is strong at visible or longer wavelengths because the atmosphere is more transparent than in the UV where Rayleigh scattered light is more prominent and therefore smooths and reduces the surface BRDF effect at UV wavelengths. Obviously, the longer the wavelength, the stronger the effects, as shown in Lorente et al. (2018) when comparing surface anisotropy effects in the near-infrared (NIR) with that in the visible.

P5, L22-25: it is not clear why the authors include the phrase about the use of both morning and afternoon MODIS sensors, since this is not 'aan de orde' in the manuscript.

Since the MODIS observations on the morning Terra overpass and afternoon Aqua overpass cover the same location with different viewing geometries, the use of data from both satellites increases the number of high quality, cloud-free multi-angle measurements collected during the satellites' 16-day repeat cycle. This reduces the uncertainty and random noise of the retrieved BRDF kernel coefficients (Schaaf et al., 2011).

We modified the text in section 2.2 (last paragraph of page 6) as follows.

Since the morning overpass (Terra) and afternoon overpass (Aqua) view the same location with different sun and viewing geometries, use of data from both satellites would double the angular samples during the 16-day repeat cycle, thus increasing the number of high quality, cloud-free observations, and reducing the uncertainty and random noise amplification of kernel coefficients retrievals (Salomon et al., 2006; Schaaf et al., 2011).

P5, L26-34: can the authors be more quantitative here and state the quantitative findings from the albedo validation exercises? Any indications for the MODIS albedo being biased low or high? What were the "accuracy requirements" exactly?

We revised that part following the reviewer's suggestion. The "accuracy requirements" for albedo for all bands in MCD43 product is 0.02 in reflectance units or 10% of surface measured values (Jin et al., 2003; Roman, et al., 2013).

We added the following to the text in section 2.2 (first paragraph of page 7).

The absolute accuracy requirement for albedo for all bands in the MCD43 product is 0.02 in reflectance units or 10% of surface-measured values (Jin et al., 2003; Roman et al., 2013). Indeed, the majority of the extensive validation campaigns on different platforms across different landscapes and seasonal cycles have demonstrated that the MCD43 product meets this requirement. These include comparisons with ground-based or airborne measurements (e.g., Wang et al., 2004 in the Tibetan Plateau; Coddington et al., 2008 over Mexico city; Wang et al., 2012 in snow-covered tundra) as well as with space-borne data (e.g., Susaki et al., 2007 in paddy fields using Advanced Spaceborne Thermal Emission and Reflection Radiometer (ASTER) and Enhanced Thematic Mapper Plus (ETM+) data; Roman et al., 2013 with Landsat and the Cloud Absorption Radiometer (CAR) data; and

Wang et al., 2014 using ETM+). However, there are a few 10 cases where MODIS retrieved albedo are smaller than field measurements, e.g., a bias of -0.01 for the visible broadband albedo (0.3-0.7µm) over FLUXNET tower sites (Cescatti et al., 2012; Wang et al., 2010).

P7, suggest to move Figure 2 to the Supplement. I think the readers can trust the experienced NASA-team to do a proper job in re-gridding, and there is no new science in here.

Figure 2 is related to pixel water fraction (1-$f_L$), an important parameter for pixels with a mixture of land and water. $f_L$ cannot be derived from OMI L1b data so we use the high-resolution, 30 arc-second static land-water mask map in MCD43 to estimate the fraction. Fig. 2 shows the need to use a high spatial resolution land/water mask to correctly estimate land/water fraction for OMI pixels on coasts and containing in-land waters. The current paper focuses only on evaluation of OMI scenes where $f_L$ =1 and this figure is here to help explain to users what this means.

We create a subsection and modified the original text as follows.

2.3 Pixel land areal fraction

The areal fraction of land (or water) for each OMI pixel is a critical parameter in TOA radiance calculation for pixels mixed with land and water (see Eq. 3). However, it cannot be estimated from OMI L1b pixel surface category flags because these binary flags do not provide information on mixed pixels. Therefore, a binary land/water classification method is developed to estimate pixel land fraction using the high-resolution, 30 arc second, static land-water mask map provided in MCD43.

First, we convert the eight surface categories from MCD43 into a binary land-water flag, merging all shorelines and ephemeral water at the MODIS spatial resolution into the land class and classifying all other water sub-categories as water. The areal fraction of land (or water) for each OMI pixel is then computed from the counts of land and water points within the OMI FOV. Typical results are shown in Figure 2.

P8, L6: please clarify what "day-1 solar irradiance spectrum" refers to. Is it the irradiance spectrum measured on 1 October 2004?

For this study, no solar irradiance is needed for GLER calculation. However, OMI LER is retrieved from OMI collection 3 data at 466 nm after normalizing the OMI radiances to one day-1 solar irradiance spectrum measured on 21 October 2004 and has been corrected to account for Earth-Sun distance.

We modified the 2$^{nd}$ paragraph of section 2.4 as follows:

Specifically, we use LER retrieved from TOA radiances at 466 nm that are computed by normalizing the OMI radiances to the OMI day-1 solar irradiance spectrum measured on 21 December 2004 along with a correction for the Earth-Sun distance when calculating OMI-derived LER.

P9, Figure 3 also appears redundant. I don't see why these (quite common) re-gridding approaches should be discussed in detail. The figure looks to me as a mere illustration of the approach described in Haines et al. [1994], so I'm afraid nothing's new here.

We understand the reviewer's point, however since reviewer #2 asked questions about our pixel gridding method in their comments, we answered them and moved the figure in question to Appendix A1 along with the relevant text.

P10, L10-12: it is unclear how application of a pseudo-spherical geometry calculation can lead to a "sphericity correction for both incoming and outgoing viewing directions". Please discuss this in more detail. How does the supposed spherical correction relate to the pseudo-spherical correction only?

We agree that some clarification is required here. For VLIDORT calculations presented in this paper, the default is to use the pseudo-spherical correction, that is to say, for multiple and single scattering calculations, solar beam attenuation (before scattering) is derived for a spherical non-refractive atmosphere. In VLIDORT, multiple scatter calculations are done for a plane-parallel medium. However, single-scattering computations are done separately, and in addition to treating solar beam attenuation in a curved atmosphere, it is also possible to treat viewing-path attenuation in a spherical atmosphere. This is what is meant by "sphericity correction for both incoming and outgoing viewing directions" - something that only applies to the single scatter computations.

We have revised the first paragraph in 2.6 GLER computation regarding use of sphericity correction as follows:

Given all necessary input parameters, TOA radiances (Icomp) are computed with the Vector Linearized Discrete Ordinate Radiative Transfer (VLIDORT) model. VLIDORT is a vector multiple scattering radiative transfer model that can simulate Stokes 4-vectors at any level in the atmosphere and for any scattering geometry with a Lambertian or non-Lambertian underlying surface (Spurr, 2006). In this study, VLIDORT computations are carried out using the pseudo-spherical correction, i.e. for both multiple and single scattering calculations, solar beam attenuation (before scattering) is treated for a spherical non-refractive atmosphere. Multiple scatter calculations are done for a plane-parallel medium. However, in the single scattering treatment, both solar-beam and line-of-sight attenuations are computed for a spherical-shell atmosphere. These "sphericity corrections" are necessary to obtain the most accurate results for geometrical configurations with large solar zenith angles, and also for wide-angle viewing scenarios. VLIDORT is executed in vector mode for our calculations, since neglect of polarization can lead to considerable errors for modeling backscattered spectra in the UV/Vis wavelength range.

P10, L19: in Eq. (2) and from the text below it is not immediately clear that Icomp

refers to the VLIDORT-simulated TOA radiance levels based on simulations with a pure Rayleigh atmosphere and the capacity of the model to account for the surface BRF. Then, R can only be found if the model can simulate I0, T, and Sb, something that VLIDORT surely can, but is not becoming clear from the text.

We agree and have made the following changes to the text, before Eq.(2):

We simulate clear sky TOA radiance (Icomp) over a non-Lambertian surface by coupling VLIDORT with the MODIS kernel-driven BRDF function (Eq. 1) from the group of analytical BRDF models available in the VLIDORT BRDF supplement to account for the surface BRDF effect on TOA radiance over land surfaces.

We also inserted the following sentence after Eq.(2):

We also computed $I_0$ , T and $S_b$ with VLIDORT by calculating TOA radiances for three values of R, and then solving three linear equations in the form of Eq. 2 to derive the three terms.

P11, L26-27: it would be appropriate to cite papers here that made the point that cloud fraction retrievals actually provide 'effective cloud' fraction information

that accounts for aerosol effects, e.g. Boersma et al. [2011], and for higher scattering in the forward direction of cloud particles, e.g. Lorente et al. [2018].

We acknowledge the reviewer's point that cloud fraction retrievals actually provide 'effective cloud' fraction information that accounts for aerosol effects, and added the references as suggested.

P11, L31: suggest to clarify that "this equation" refers to Eq. (2).

We made this change as the reviewer suggested.

P13, L1-5: can you provide a quantitative statement on how much OMI LER is typically higher than GLER? From the intercepts one would say the difference is 0.01, but possibly the mean or median difference is a more meaningful metric. I would also encourage if the authors could report whether there is a pattern in how the LER-GLER differences change between different regions/surface types.

We have added Table 3 showing the mean difference between GLER and OMI-derived LER along and the $r^2$ values for the regions analyzed in the paper. We believe this more effectively and efficiently conveys the information we presented previously in multiple scatter plots. We note that due to relaxation of our cloud screening criteria, as described in the appendix, the mean differences between are ~0.01-0.02. The median differences are close to the mean (at most 0.001-0.002 median/mean difference for data shown in Table 3). However, we have found that the mode of the difference is smaller than the mean (up to 0.008) and may be more representative of the true bias, as shown in the appendix. Because we separate our analysis by season, we have seasons with little data, making the mode difficult to calculate. For simplicity we elect to report the mean values in the new Table 3. We note that the mode is smaller than the mean, so we are reporting the larger of the two possible estimations of the bias.

P15, L12-14: the hypothesis that localized floodplains darken after rain resulting in a signal detected by OMI LER (daily data), but not by GLER (MODIS-based 8-day data) needs to be substantiated. It sounds possible, but there is no basis for this statement from a result shown.

We examined MODIS data for outliers where GLER is significantly higher than the OMI-derived LER and found several examples associated with two ephemeral lakes which only retain water for very short periods: Salar de Uyuni in southwest Bolivia and Lake Frome in South Australia. Since MCD43GF is derived by fitting

MODIS observations within a 16-day period, rapid changes due to flooding of these surfaces is not well captured. We made the following changes in the text:

These data are from the Salar de Uyuni salt lake in southwest Bolivia and Lake Frome in South Australia, which only flood during heavy rain events. These basins typically retain water for short periods of time and likely would not be captured in the 16 day MODIS BRDF data (Schaaf et al. 2011).

P16, L5-7: it is possible that OMI data is indeed affected by residual clouds or aerosols leading to higher reflectances. But it is also possible that the MODIS-based data have been overcorrected for atmospheric effects. As long as no evidence is presented to obtain confidence in the validity of the atmospheric correction (and data screening) applied to the MDC43 suite, we cannot know if it's one or the other, see my main concern.

As discussed in response to point 1a above, we examined the literature on the MODIS atmospheric correction algorithm applied in the MOD09 product that is upstream of MCD43. The MODIS atmospheric correction algorithm has been evaluated extensively and we found no evidence of errors large enough to make the correction or screening the primary source of the bias. This does not rule out the possibility that some portion of the bias is due data screening or atmospheric correction, but we think residual clouds or aerosols, and perhaps a contribution from calibration bias, are more likely causes.

P19, L8-10: please explain how using the GLER reduces the tropospheric AMF. Is it via the increased cloud fractions (more screening), or the lower clear-sky AMFs because of the darker surface, or both?

Via both, because both clear-sky and cloudy AMFs decrease as surface reflectivity decreases. That means the clear-sky tropospheric AMF would be smaller when GLER is smaller than the climatological LER data, as we generally find is the case for the Kleipool climatology (see Vasilkov et al., 2017).

P19, L20: please clarify how differences in calibration could explain the bias of 0.01 between OMI LER and GLER. Is there any reason to believe that OMI is calibrated such that it detects too low, or MODIS too high reflectances? Have level-1 data been compared in the first place?

This is an important issue, and we are grateful to both reviewers for drawing attention to it. We have added information on the uncertainties of the MODIS

instruments to section 2.2.

The calibration uncertainty for MODIS band 3 is within 2% (Xiong et al., 2005). The MODIS Aqua solar reflective bands including band 3 were corrected for a time-dependent drift in Collection 5 (Wu et al., 2013) but errors in MODIS Terra of up to 5% across the scan developed approximately 5 years after launch and this error was not sufficiently corrected in Collection 5 (Sun et al., 2014; Lyapustin et al., 2014).

We also added information on what is known of the relative calibration of the two instruments, and the effect of calibration error is on GLER and LER differences in new paragraphs in the section 4 (Discussion).

In addition to background non-absorbing aerosol and/or residual cloud contamination, it is important to consider that the GLER-LER bias may be due in part to differences in the MODIS and OMI radiance calibration. Sensitivity analysis of Eq. 2 used to compute LER and GLER shows that a 1% error in TOA radiances will produce errors in LER of up to 0.003 in surface reflectivity. A bias of 0.01 between GLER and LER requires a difference in MODIS and OMI TOA radiance of at least 3% for brighter land scenes (LER >=0.2), and differences of 10% for darker land scenes (LER <= 0.05). MODIS TOA radiances would thus have to be 3-6% low relative to OMI to explain the bias seen in GLER-LER for bright scenes, and 10-20% low for dark scenes.

Jaross and Warner (2008) compared TOA reflectances from OMI and MODIS with radiative transfer model simulations over Antarctica, accounting for the BRDF of the snow surface. By indirect comparison, OMI Collection 3 and MODIS Collection 5 agreed to within 1% at the start of the OMI mission. They estimated the uncertainty of their technique is 2%. This level of disagreement is smaller than needed to explain all of the 0.01-0.02 bias of GLER over dark scenes. We therefore conclude that only some of the bias can be attributed to calibration differences. Additional information about the relative calibration of OMI and MODIS is provided in Appendix D.

Relative sensor drift is also a concern in comparing the GLER product using the MODIS calibration with LER from OMI. Aqua MODIS appears to be well corrected in Collection 5 but the MCD43 product also uses data from the Terra instrument, which has degraded appreciably over the lifetime of the mission. However, we

find no evidence of time dependent change in Collection 5 MODIS BRDF data. We suspect the time-dependent and scan angle-dependent error in the Collection 5 MODIS Terra calibration data have somehow been avoided. Since OMI drift has not been fully corrected, and the MODIS drift has been removed (or avoided in the case of Terra, apparently) the slight decrease of OMI LER relative to GLER between 2006 and 2015 in figure 8 may be due to the 1-1.5% calibration drift in OMI radiances.

We have also added Appendix D, Relative calibration of OMI and MODIS, to provide additional information about the relative calibration of the level 1b data from the two instruments used in this study.

---

## Author Comment (AC2) · 7 May 2019

Dear Editor,

We thank the reviewer for the evaluation of our paper and useful comments that helped to improve the manuscript. Below are our responses to each comment. Reviewer's comments are in blue, the responses are in black; the text added to the manuscript is in red.

On behalf of the authors,

Wenhan Qin

**Responses to comments from Referee #2:**

(1)  The paper provides some explanations for the differences that are found between the MODIS-based GLER and the OMI-derived LER. These explanations are neglect of aerosol in the simulations, the possibility of cloud contamination in the OMI observations, and calibration differences between the MODIS and OMI instruments.

The reviewer is correct that in the simulations, we did not consider the aerosol or cloud as it is stated in the last paragraph of section 2.4 (page 9): aerosols are not included in the VLIDORT simulation of TOA radiances (from which GLER is computed). So, the GLER-LER comparison screened OMI data to minimize impacts of absorbing aerosols and clouds.

We realized even after applying proper criteria of aerosol index (AI) <0.5 and effective cloud fraction (ECF)<0.04 for data screening, the GLER- LER difference is still subject to the impact from background aerosols (mostly non-absorbing) and residual clouds, as we clarified in section 3.1. However, current generation of the satellite cloud and trace gas retrieval algorithms do not treat aerosols explicitly and this analysis is beyond the scope of this paper.

We agree that the impact of the Level 1b calibration differences between OMI and MODIS needs to be addressed, and we added more details in the updated manuscript (see detailed response below).

What is actually known about the calibration of MODIS compared to that of OMI? Haven't there been any studies comparing the two instruments? What are the

calibration differences needed to explain the systematic differences between the MODIS-based GLER and the OMI-derived LER?

This is an important issue, and we are grateful to both reviewers for drawing attention to it. We have added information on the uncertainties of the MODIS instruments to section 2.2.

The calibration uncertainty for MODIS band 3 is within 2% (Xiong et al., 2005). The MODIS Aqua solar reflective bands including band 3 were corrected for a time-dependent drift in Collection 5 (Wu et al., 2013) but errors in MODIS Terra of up to 5% across the scan developed approximately 5 years after launch and this error was not sufficiently corrected in Collection 5 (Sun et al., 2014; Lyapustin et al., 2014).

We also added information on what is known of the relative calibration of the two instruments, and the effect of calibration error is on GLER and LER differences in new paragraphs in the section 4 (Discussion).

In addition to background non-absorbing aerosol and/or residual cloud contamination, it is important to consider that the GLER-LER bias may be due in part to differences in the MODIS and OMI radiance calibration. Sensitivity analysis of Eq. 2 used to compute LER and GLER shows that a 1% error in TOA radiances will produce errors in LER of up to 0.003 in surface reflectivity. A bias of 0.01 between GLER and LER requires a difference in MODIS and OMI TOA radiance of at least 3% for brighter land scenes (LER >=0.2), and differences of 10% for darker land scenes (LER <= 0.05). MODIS TOA radiances would thus have to be 3-6% low relative to OMI to explain the bias seen in GLER-LER for bright scenes, and 10-20% low for dark scenes.

Jaross and Warner (2008) compared TOA reflectances from OMI and MODIS with radiative transfer model simulations over Antarctica, accounting for the BRDF of the snow surface. By indirect comparison, OMI Collection 3 and MODIS Collection 5 agreed to within 1% at the start of the OMI mission. They estimated the uncertainty of their technique is 2%. This level of disagreement is smaller than needed to explain all of the 0.01-0.02 bias of GLER over dark scenes. We therefore conclude that only some of the bias can be attributed to calibration differences. Additional information about the relative calibration of OMI and MODIS is provided in Appendix D.

Relative sensor drift is also a concern in comparing the GLER product using the MODIS calibration with LER from OMI. Aqua MODIS appears to be well corrected in Collection 5 but the MCD43 product also uses data from the Terra instrument, which has degraded appreciably over the lifetime of the mission. However, we find no evidence of time dependent change in Collection 5 MODIS BRDF data. We suspect the time-dependent and scan angle-dependent error in the Collection 5 MODIS Terra calibration data have somehow been avoided. Since OMI drift has not been fully corrected, and the MODIS drift has been removed (or avoided in the case of Terra, apparently) the slight decrease of OMI LER relative to GLER between 2006 and 2015 in figure 8 may be due to the 1-1.5% calibration drift in OMI radiances.

We have also added Appendix D, Relative calibration of OMI and MODIS, to provide additional information about the relative calibration of the level 1b data from the two instruments used in this study.

For the GLER no OMI measurement data are used. Only the OMI ground pixel extent and geolocation are used and MODIS BRDF data are then used to calculate the GLER for that OMI pixel. So, the MODIS-derived GLER, which is to be used for OMI retrievals, inherited the calibration differences that exist between MODIS and OMI.

That is correct. The MODIS-based GLER inherits the MODIS calibration uncertainty, and when is used for OMI retrievals, it introduces the calibration differences between MODIS and OMI Level 1b data.

For the GLER calculation, not only the OMI ground pixel extent and geolocation are used, but the OMI geometries (solar, viewing and relative azimuth angles) are needed because GLER depends on all these angles.

It might be good to mention (more explicitly) in the paper that using the GLER for OMI-based retrievals can result in the introduction of calibration inconsistencies in these retrievals.

Thank you for pointing this out. We added the following into the 3rd paragraph of the section 4 (Discussion).

Sensitivity analysis of Eq. 2 used to compute LER and GLER shows that a 1% error in TOA radiances will produce errors in LER of up to 0.003 dependent in the surface reflectivity. A bias of 0.01 between GLER and LER requires a difference in

MODIS and OMI TOA radiance of at least 3% for brighter land scenes (LER >= 0.2), and differences of 10% for darker land scenes (LER <= 0.05). MODIS TOA radiances would thus have to be 3-6% low relative to OMI to explain the bias seen in GLER-LER for bright scenes, and 10-20% low for dark scenes.

(2) What are, to your knowledge, the expected differences between MODIS-derived GLER and the MODIS BRDF? In other words, what are the errors the user would introduce if he/she would use the MODIS-derived GLER as if it would be a BRDF?

GLER is calculated by exactly matching OMI-measured and calculated TOA radiances over a non-Lambertian surface as described in Eq.2. However, the average photon path lengths for the measured and calculated signals can still be different and the larger the viewing zenith angle, the larger the difference. Our simulations have shown the difference between GLER and BRF does not exceed 0.01 at SZA of 45° and viewing zenith angle < 70° within the solar-OMI principal plane. Vasilkov et al. (2017) reports the simplification of using the GLER approach relative to the full BRDF treatment can lead to small biases in the calculation of AMFs of up to 6% (see Fig. 10a) and similar biases in the retrieved NO2 vertical columns.

Is the 0.8% mentioned in section 2.1, page 5, lines 18-19 a representative percentage for this? Is this 0.8% valid for 466 nm?

The referenced value of the error 0.8% due to Lambertian assumption does look small compared to the current literature. Although this result is for band 1 (645 nm), a similar result is expected for band 3 (466nm) which we use in this study, because for land surfaces other than deserts the surface reflectance in these two bands is similar.

Please note that we removed this statement in our revision because other independent studies have shown different numbers for the Lambertian assumption. More importantly, the uncertainty in our GLER product is tied more to uncertainty of MCD43 data than MOD09, which the above number pertains to.

Does the term "green band" in line 20 refer to the 555-nm MODIS band (band 4)?

Yes, this is correct: "green band" for MODIS is band 4 (550 nm) in their terminology.

(3) The paper introduces GLER calculated at 466 nm. For this, MODIS BRDF information is used from MODIS band 3. However, MODIS band 3 is centered around 470 nm, and not around 466 nm. On page 7, lines 6-7, an explanation for the 4-nm difference is given which suggests that the 470-nm MODIS BRDF is just used "as is" at 466 nm

If this is the case, wouldn't it be better to say that the GLER is representative for 470 nm(and not for 466 nm, even though the GLER retrieval is performed at this wavelength)? What are the technical reasons for doing the GLER retrieval at 466 nm and not at 470nm? Is it because of absorption by trace gases at 470 nm? If so, which trace gases?

MODIS band 3 has a 20 nm bandwidth (459-479 nm) centered near 470 nm. The land surface reflectivity does not have high-frequency spectral structure within this 20 nm range so wavelengths in this range, such as 466 nm, will be representative of the band in terms of land surface reflectance.

However, the same is not true for the atmospheric effects. The reason we picked 466 nm for the GLER calculation is because (1) 466 nm is further away from the $O_2$-$O_2$ absorption centered around 477 nm, and from an ozone absorption feature near 470 nm, and (2) 466 nm is the wavelength used in our cloud algorithm to retrieve effective cloud fraction (ECF) (Vasilkov et al., 2017). Observations at 466 nm are relatively free of atmospheric inelastic, rotational-Raman scattering (RRS) as well, which is important because OMI has narrow band pass and thus is sensitive to the RRS errors.

We modified the last paragraph of section 2.2 as follows.

Since kernel coefficients depend on wavelength, for the present study we selected MODIS band 3, the shortest wavelength in the MCD43GF product, with a center wavelength of 470 nm (ranging from 459 to 479 nm) to represent 466 nm, which is the wavelength used in our cloud algorithm to retrieve effective cloud fraction (ECF) (Vasilkov et al., 2017). Observations at this wavelength are relatively free of atmospheric rotational-Raman scattering (RRS) and trace gas absorption.

(4) section 1, page 3, line 9:

Here the paper mentions a few papers on LER databases. Could you also mention the databases/instruments behind these references?

Please also add the (more recent) reference to Tilstra et al. (2017), about LER retrieval from the GOME-2 instrument. https://doi.org/10.1002/2016JD025940

We appreciate the reviewer pointing out these LER datasets from GOME/2 and SCIAMACHY. We added references to those databases as well as the recent publication as suggested.

We also added the following to the Introduction:

For example, Herman and Celarier (1997) from the Total Ozone Mapping Spectrometer (TOMS) at 340 and 380 nm, Koelemeijer et al. (2001) from the Global Ozone Monitoring Experiment (GOME) in 11 wavelengths between 335-772 nm, Kleipool et al. (2008) from OMI in 23 wavelengths at 328-499 nm, and more recently Tilstra et al. (2017) from GOME-2 in 21 wavelengths between 335-772 nm as well as from the Scanning Imaging Absorption Spectrometer for Atmospheric Chartography (SCIAMACHY) in 29 wavelengths from 335-1670 nm. These climatologies are constructed by computing statistical values representative of multiple years of observations made with different sun-satellite viewing geometries. In order to minimize cloud contamination, they may be based on a lower percentile (e.g., Herman and Celarier, 1997) and/or the mode of the LER histogram depending on surface type (e.g., Koelemeijer et al., 2001; Kleipool et al., 2008; Tilstra et al., 2017).

 (5) section 2.3/Fig. 3:

When explaining how the point-in-polygon method is used, please mention briefly that the "real" OMI pixel is not rectangular, and that this fact alone already can lead to (small) differences between the (MODIS-derived) GLER and the OMI-derived LER.

We agree with this point made by the reviewer and we moved this to the Appendix A1 with more information describing our collocation methodology, which now says the following:

The collocation methodology is shown schematically in Fig. 12. The OMI pixel is first defined from the four ground pixel corner points provided in the OMPIXCOR data product as a 4-sided polygon. A sample space is then constructed along constant latitudinal boundaries, with the corner points tangent to the boundaries of the sample space as shown. All pixels from the MODIS BRDF/Albedo product and ancillary data sets inside the sample space are tested using the point-inpolygon method (Haines, 1994). For this application, we used the corner points for the VIS channel corresponding to 75% of the energy in the along-track field of view. This definition assumes the pixels across the track share boundaries with their two adjacent neighbors (except for the pixels at the far edge of the swath), while the pixels along the track of the satellite overlap (reference to OMPIXCOR Readme).  De Graff et al. (2016) showed the actual shape of the OMI pixel is not exactly a rectangular polygon but rather is best represented by a super Gaussian. They also showed that the optimal overlap function between OMI and MODIS depends on the scene and the time difference between the satellites. We do not consider these factors as critical to this application because the GLER is based on MCD43GF, an 8-day gridded MODIS BRDF product from Terra and Aqua. Small errors in the pixel shape should only have a minimal impact on our results.

Perhaps you could refer to the paper by De Graaf et al. (2016) about the size and shape of the OMI pixels. https://doi.org/10.5194/amt-9-3607-2016

We thank the reviewer to point this out. Indeed, the actual shape of the OMI pixel is not exactly a rectangular polygon but rather is best represented by a super Gaussian. We added the reference De Graff et al. (2016) as suggested.

In this paper, OMI and MODIS (band 3) reflectances are compared to each other using different sizes and shapes for the OMI point-spread function.

This is correct, and so we have to collocate MODIS pixels at 30 arc-second resolution within the OMI pixels as described in Appendix A1.

(6) section 3.2, page 13, line 16:

The paper mentions here that surface BRDF does not change on a day-to-day basis. But this can happen in certain cases, as explained in section 3.3, page 15, lines 12-14. Perhaps you could change the sentence to "While surface BRDF in general does not ..." (or something similar)?

We agree with this comment and have made minor changes to this sentence in section 3.2.

Surface BRDF or albedo change is small on a day-to-day basis, with the exception of extreme events such as fires and floods that are not captured with the 16-day MODIS dataset. There is, however, noticeable variability in the BRDF and albedo between seasons due to land cover changes throughout the year.

 (7) In the paper the GLER and the OMI-derived LER are also compared to the Kleipool climatology. Which field is taken from the Kleipool climatology? Is it the field "MonthlyMinimumSurfaceReflectance" or is it the field "MonthlySurfaceReflectance"? If it is the "MonthlySurfaceReflectance" field, then that would probably explain part of the higher values of the Kleipool climatology compared to the OMI-derived LER for at least some of the land cover types in Figure 9. In fact, I think that in these analyses it would be better to use the traditional "MonthlyMinimumSurfaceReflectance" field.

In any case, it would be good to mention in the paper which of the two fields was used in the analyses.

The field "MonthlySurfaceReflectance" is used because this is what was used in our NO2 and O2-O2 cloud algorithms before switching to GLER. But we acknowledge this is a common point of confusion, so for comparison purposes, we include data from both the "Monthly Surface Reflectance" and "Monthly Minimum Surface Reflectance" fields in figures in our revised manuscript.

(8) small typo in a reference, page 27, line 20:

Haines, E.,: –> Haines, E.:

Thank you for pointing this out this typo.  This has been corrected in the revised version.